# Learn from Interactions: General-Sum Interactive Inverse Reinforcement Learning

## Abstract

This paper studies the problem that a learner aims to learn the reward function of the expert from the interactions with the expert and how to interact with the expert. We formulate the problem as a stochastic bi-level optimization problem and develop a double-loop algorithm "general-sum interactive inverse reinforcement learning" (GSIIRL). In the GSIIRL, the learner first learns the reward function of the expert in the inner loop and then learns how to interact with the expert in the outer loop. We theoretically prove the convergence of our algorithm and validate our algorithm through simulations.

## 1 Introduction

Inverse reinforcement learning (IRL) has been widely applied to various domains including robotics (Ziebart et al., 2008; Okal & Arras, 2016) and cyber security (Zhang et al., 2019; Elnaggar & Bezzo, 2018). In IRL, a learner aims to learn a reward function and an associated policy that are consistent with the demonstrations of an expert. IRL focuses on learning the reward function since "the reward function is the most succinct, robust, and transferable definition of the task" (Ng & Russell, 2000). When the reward function is learned, methods like reinforcement learning (RL) can be used to learn the corresponding policy.

In this paper, we consider a variant problem of IRL called interactive IRL. In the classic IRL, the learner only passively observes the experts and learns from demonstrations in isolation from the experts, so that the learner does not affect the expert's demonstrated trajectories. In interactive IRL, the learner actively interacts with the expert and thus can influence the trajectories of the expert. That is, the learner is not only an observer but also a participant. Consider a scenario where a human (i.e., expert) and a ground mobile robot (i.e., learner) navigate in a 2D plane while their destinations are distinct. The human usually has a higher priority so the robot needs to bypass or slow down for the human to cross. Meanwhile, the robot also wants to reach its destination as soon as possible. Therefore, the interaction between the robot and the human is a general-sum game. Notice that the robot has a human-independent goal (i.e., its own destination) and a human-dependent goal (i.e., bypassing the human). The reward function of the robot can be decomposed into two parts and each part captures one of the goals. For example, the reward function in (El-Shamouty et al., 2020) is decomposed into the distance of the robot to its goal location and the fact whether a collision between the robot and the human happens. Motivated by the above example, this paper considers the general-sum game where the learner's reward function is decomposed into the expert-dependent and the expert-independent parts. The learner knows the expert-independent reward function and is unaware of the expert-dependent reward function and the expert's reward function. The learner aims to learn the expert reward from expert demonstrations during the interactions. The benefits of learning the expert's reward include facilitating the learner's policy learning and understanding a transferable expert model. When the transition probabilities of a Markov game (MG) substantially change, the policy needs to be re-trained, but the reward function could be the same. Fully cooperative (Palaniappan et al., 2017; Büning et al., 2022; Kamalaruban et al., 2019; Hadfield-Menell et al., 2016) and fully competitive (Zhang et al., 2019; Wang & Klabjan, 2018; Goktas et al.) interactive IRL problems are the special cases of our interactive IRL problem. The fully cooperative interactive IRL problems assume the learner and the expert cooperate with each other and the reward functions of the learner and the expert are identical. The fully competitive interactive IRL solves a zero-sum game where the reward functions of the learner and the expert are opposite.

Our interactive IRL problem is relevant to classic IRL, multi-agent IRL (MA-IRL) (Lin et al., 2019; Yu et al., 2019; Liu & Zhu, 2022), and multi-agent RL (MARL) (Lowe et al., 2017; Yu et al., 2022; Kuba et al., 2021). In the classic IRL and MA-IRL, the learners are isolated from the experts. However, the learner actively influences the trajectories of the expert in our interactive IRL problem. As a result, classic IRL and MA-IRL can not be directly applied to our interactive IRL problem. MARL is usually utilized to learn the policies with the given rewards. it can not used to learn the reward function of the expert. Although the model-based MARL (Moerland et al., 2023) can learn the reward function of the learner, it still can not learn the reward function of the expert. Therefore, MARL can not solve our interactive IRL problem.

**Contribution.** The contributions of this paper are four-fold. First, we study the general-sum interactive IRL problem, which includes fully cooperative and fully competitive IRL as special cases. We formulate the problem as a stochastic bi-level optimization problem where the lower level learns a reward function of the expert through IRL and the upper level learns a reward function for the learner and a corresponding policy through RL. Second, we develop a double-loop algorithm "general-sum interactive inverse reinforcement learning" (GSIIRL) to solve the bi-level optimization problem. The outer loop solves the upper-level optimization problem and the inner loop solves the lower-level optimization problem. In both the inner loop and the outer loop, we use the projected stochastic gradient descent (SGD) to update the decision variables. In particular, we leverage the simultaneous perturbation stochastic approximation (SPSA) (Spall, 1992) to approximate the hypergradients in the outer loop since the hypergradients include the Hessian of the lower-level objective function. With the SPSA, the computational complexity per iteration of the outer loop reduces from $\mathcal{O}(T^3)$ to $\mathcal{O}(T^2)$ where $T$ is the maximum length of the trajectories. Third, we show that the expected hypergradients decrease at the rate of $\mathcal{O}(\frac{1}{\sqrt{K}})$ where $K$ is the iteration number of the outer loop. Fourth, we validate our algorithm through three experiments.

## 2    RELATED WORK

**IRL and Multiagent IRL.** As (Ng & Russell, 2000) mentioned, IRL faces the challenge that one demonstration can be explained by multiple reward functions. A number of methods have been developed to address the challenge. The maximum margin method in (Ng & Russell, 2000) aims to maximize the differences between the reward of the optimal action and that of the second-best action. Feature expectation matching in Abbeel & Ng (2004) aims to minimize the difference between the feature expectation of a learned policy and the empirical feature of demonstrations. Based on the previous idea, maximum entropy IRL (ME-IRL) (Ziebart et al., 2008) finds the reward function by maximizing the entropy while imposing the feature expectation matching as a constraint.

The aforementioned works are only focused on a single expert. When multiple experts interact with each other in the environment, it becomes MA-IRL. To learn the reward functions of all experts, papers (Lin et al., 2019; Yu et al., 2019; Liu & Zhu, 2022) extend the above single-expert IRL to multi-expert IRL where a learner or a group of learners recover the reward functions of all experts from their demonstrations. Consider a MG which is controlled by two experts. Each expert knows its reward function which depends on the MG state and/or the joint action of the experts. Since they know the reward functions, the experts can execute the policies that maximize the reward functions and provide the induced trajectories of the MG as demonstrations. In these papers, the learners only passively observe the experts and learn from demonstrations in isolation from the experts. The learner cannot influence the MG but only observes the demonstrations.

**Fully cooperative and fully competitive interactive IRL.** Recent papers (Palaniappan et al., 2017; Büning et al., 2022; Kamalaruban et al., 2019; Hadfield-Menell et al., 2016) study a fully cooperative setting where the learner has the same reward function as the expert. The learner learns this shared reward function. The papers formulate the interactive IRL problem as a Stackelberg game where the lower level learns the policy of the expert and the upper level finds the reward function of the expert. Papers (Zhang et al., 2019; Wang & Klabjan, 2018; Goktas et al.) study a fully competitive setting where the reward function of the learner is the opposite of that of the expert and formulate the problem as a zero-sum game. The learner learns the opposite of the expert's reward function. These interactive IRL methods impose strong assumptions on the relationship between the expert's reward function and the learner's reward function.

In this paper, the relationship between the reward functions of the learner and the expert is arbitrary. The fully cooperative and fully competitive cases are special cases of the above scenario. Let us ignore the expert-independent reward function. If the expert-dependent reward function is identical to the expert's reward function, it reduces to the fully cooperative case. If the expert-dependent reward function and the expert's reward function are opposite, it reduces to the fully competitive case.

**Bi-level optimization.** Bi-level optimization has been applied to many machine learning problems, including meta-learning (Lee et al., 2019; Xu & Zhu, 2022), hyperparameter optimization (Pedregosa, 2016; Xu & Zhu, 2023), and IRL (Liu & Zhu, 2022). The descent method (Kolstad & Lasdon, 1990; Xu & Zhu, 2023) is a classic approach for solving bi-level optimization problems. This method requires to calculate the second-order Hessian of the lower-level objective function. To avoid computing the Hessian, a common way is to use the finite difference method (Strikwerda, 2004) to approximate the Hessian. This paper adopts SPSA to further reduce the number of measurements by disturbing all directions of the decision variable at one time, so it only requires 2 measurements of the objective function.

## 3 MOTIVATING EXAMPLE

Consider the scenario of a human and a robot moving and coordinating in a 2D plane.

**Human model.** The human moves in the 2D plane by changing his/her velocity $v_h$ and wants to reach his/her destination $g_h$ as soon as possible. This goal is captured by the reward function $-\|g_h - p_h\|$ where $p_h$ is the position of the human. The human can measure $p_h$ and assess the reward function. When interacting with the robot, the human exhibits different emotions. For example, the human is angry when the robot is close or he/she is slow, the human is happy when he/she moves fast (Bera et al., 2019).

**Robot model.** Analogous to the human, the robot decides the velocity $v_r$ to reach its destination $g_r$ as soon as possible. The robot can measure its current position $p_r$ and access the reward function $-\|g_r - p_r\|_2$. In addition, the robot aims to keep the human with positive emotions like happiness during the interactions. This goal of the robot is captured by the reward function $h(p_r, p_h, v_r, v_h)$. The function $h$ is unknown to the robot but the robot can measure the value of $h(p_r, p_h, v_r, v_h)$ given any locations $p_r, p_h$ and velocities $v_r, v_h$. For example, the robot captures the facial expressions of the human through a camera and recognizes his/her emotions using a deep neural network-based classifier. Positive emotions like happiness are interpreted as positive values and negative emotions like anger are interpreted as negative values (Arakawa et al., 2018). These are the measured values of $h(p_r, p_h, v_r, v_h)$. The overall reward function of the robot is given by $-\|g_r - p_r\|_2 + h(p_r, p_h, v_r, v_h)$ where the second term is dependent on the human but the first term is not.

**Robot learning.** The robot aims to learn the expert's reward function $-\|g_h - p_h\|$ from observed positions and velocities of the human.

## 4 MODEL AND PROBLEM STATEMENT

In this section, we will introduce the interactive IRL problem, which generalizes the example in Section 3.

**Markov Game model.** We model the interactions between a learner and an expert based on a finite horizon MG $(S, A, P, T, r_{ld} + r_{li}, r_e, \gamma)$. The elements of the MG are defined as follows.

- $S \triangleq S_l \times S_e$ is a non-empty Borel state space where $S_l$ and $S_e$ are associated with the learner and the expert, respectively; $s \in S, s_l \in S_l, s_e \in S_e$.
- $A \triangleq A_l \times A_e$ where $A_l$ and $A_e$ are non-empty Borel spaces of actions associated with the learner and the expert, respectively; $a \in A, a_l \in A_l, a_e \in A_e$.
- $P(s'|s, a)$ is the probability density for the state transition from $s$ to $s'$ by taking joint action $a$.
- $T$ is the finite time horizon.
- $r_{ld} + r_{li}$ is the reward function of the learner. It includes two parts: the expert-dependent reward function $r_{ld}(s, a)$ and the expert-independent reward function $r_{li}(s_l, a_l)$. The func-

tion $r_{ld}$ maps the state-action pairs $(s, a)$ to bounded rewards, and the function $r_{li}$ maps the learner's state-action pairs $(s_l, a_l)$ to bounded rewards.

- $r_e$ is the reward function of the expert which maps the state-action pairs $(s, a)$ to bounded rewards.
- $\gamma \in (0, 1)$ is the discount factor.

Referring to the motivating example in Section 3, the robot is the learner and the human is the expert. The human-dependent reward function $r_{ld}$ is $h(p_r, p_h, v_r, v_h)$, the human-independent reward function $r_{li}$ is $-\|g_r - p_r\|_2$ and $r_e$ is the reward function $-\|g_h - p_h\|_2$ of the human .

Define $\pi_l(a_l|s)$ as the policy of the learner, $\pi_e(a_e|s)$ as the policy of the expert and $\pi(a|s) \triangleq \pi_l(a_l|s) \times \pi_e(a_e|s)$ as the joint policy. The joint policy $\pi$ represents the probability density for the learner choosing $a_l$ and the expert choosing $a_e$ at state $s$. When the policy $\pi$ is executed, the MG generates a trajectory $\zeta = s^0, a^0, s^1, a^1, \cdots, s^{(T-1)}, a^{(T-1)}$.

The goal of the learner is to find a policy $\pi_l$ such that the discounted cumulative reward $E^{\pi_l, \pi_e}[\sum_{t=0}^{T-1} \gamma^t(r_{ld}(s^t, a^t) + r_{li}(s_l^t, a_l^t))]$ is maximized. Analogously, the expert aims to find a policy $\pi_e$ which maximizes $E^{\pi_l, \pi_e}[\sum_{t=0}^{T-1} \gamma^t r_e(s^t, a^t)]$. The optimal policy $\pi^*$ is the Nash equilibrium of the MG, and such Nash equilibrium exists according to Theorem 2 in (Nowak, 2003). Notice that the MG is a framework for MARL, many MARL algorithms (Lowe et al., 2017; Yu et al., 2022; Kuba et al., 2021) have been developed to approximate the Nash equilibrium. In this paper, we assume that $\pi^*$ is obtained by some existing MARL algorithms.

**Knowledge of the learner.** The learner does not know $r_{ld}$ and $r_e$ but knows other elements $(S, A, P, T, \gamma)$ of the MG. Although the reward function $r_{ld}$ is unknown, the learner can access the value of $r_{ld}(s, a)$ given any state-action pair $(s, a)$. During the interactions, the learner can observe the trajectories of the MG. Referring to the robot in Section 3, it can notice the value of $h(p_r, p_h, v_r, v_h)$ from the emotions of the human and observe the positions and velocities of the robot and the human during the interactions.

**Goal of the learner.** Using the trajectories of the MG, the learner aims to learn $r_e$.

**Related problems.** The aforementioned MG is a general-sum game between the learner and the expert. For the special case with $r_{ld}(s, a) = r_e(s, a), \forall(s, a) \in S \times A$ and $r_{li}(s_l, a_l) = 0, \forall(s_l, a_l) \in S_l \times A_l$, the reward functions of the learner and the expert are identical and this case reduces to the fully cooperative interactive IRL in (Palaniappan et al., 2017; Büning et al., 2022; Kamalaruban et al., 2019; Hadfield-Menell et al., 2016). When $r_{ld}(s, a) = -r_e(s, a)$ , $\forall(s, a) \in S \times A$ and $r_{li}(s_l, a_l) = 0, \forall(s_l, a_l) \in S_l \times A_l$, the MG reduces to the fully competitive interactive IRL in (Zhang et al., 2019; Wang & Klabjan, 2018). In classic IRL (Ziebart et al., 2008; Abbeel & Ng, 2004; Ziebart et al., 2010; Lin et al., 2019; Liu & Zhu, 2022), the learner only passively observes the demonstrations of the expert and does not influence the expert. In contrast, the learner in our MG actively interacts with the expert. Consider the case where the MG is independent of the learner, i.e., the state transitions and the reward function of the expert are independent of the action of the learner. The case reduces to the classic IRL.

As $\pi_e$ is influenced by $\pi_l$, the learner actively influences the trajectories of the expert. We can not directly utilize classic IRL and MA-IRL for our interactive IRL problem. MARL is unable to solve our interactive IRL problem because the learner learns $\pi^*$ instead of $r_e$ through MARL.

## 5 PROBLEM FORMULATION

The learner aims to learn both $r_e$ and $r_{ld}$. Since $r_{ld}$ is unknown to the learner, it cannot apply algorithms for classic IRL to solve the learning problem. In this section, we formulate the learning problem as a bi-level optimization problem where the learner makes the first move to maximize its reward function and the expert responds accordingly to maximize its own reward function.

**Learning expert's reward function $r_e$.** Recall that $r_e$ is unknown to the learner. The learner uses the parametric function $r_e^{\theta_e}$ to estimate $r_e$ where $\theta_e \in \Theta_e \triangleq \{\theta_e | \|\theta_e\| \leq 1\}$. When the ground truth of $r_{ld}$ is known to the learner, the trajectories are sampled from the MG $(S, A, P, T, r_{ld} + r_{li}, r_e, \gamma)$ with its optimal policy $\pi^{r_{ld}, r_e}$ corresponding $r_{ld} + r_{li}$ and $r_e$. However, the learner is unaware of $r_{ld}$. The learner estimates $r_{ld}$ using $r_{ld}^{\theta_l}$ where $\theta_l \in \Theta_l \triangleq \{\theta_l | \|\theta_l\| \leq 1\}$, and aims to learn the ex-

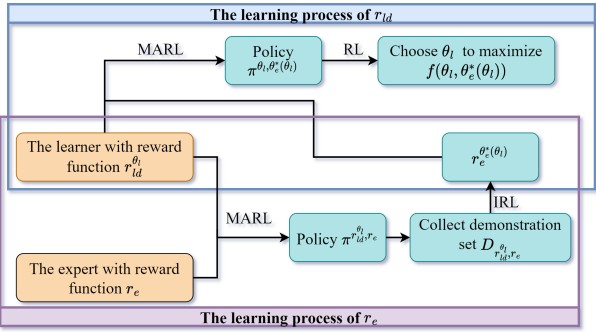

Figure 1: Flow chart of the overall learning process. The process of learning $r_{ld}$ is included in the upper block and the process of learning $r_e$ is included in the lower block.

pert reward function under any given estimate $r_{ld}^{\theta_l}$. Given $r_{ld}^{\theta_l}$, the learner collects the trajectories set $D_{r_{ld}^{\theta_l}, r_e}$ under the policy $\pi^{r_{ld}^{\theta_l}, r_e}$ which is the optimal policy of the MG $(S, A, P, T, r_{ld}^{\theta_l} + r_{li}, r_e, \gamma)$. Since the expert adopts the ground-truth reward function $r_e$, the trajectories recorded from the interactions are the demonstrations of the expert. As $r_{ld}^{\theta_l}$ is known to the learner, learning $r_e$ from the demonstrations is a special case of the standard two-agent IRL problem (Lin et al., 2019; Yu et al., 2019; Liu & Zhu, 2022) with one of the reward functions known to the learner. The learning process is shown in the upper block of Figure 1 and formulated as the following ML-IRL problem:

$$\theta_e^*(\theta_l) = \underset{\theta_e \in \Theta_e}{\arg\min} \quad L(\theta_l, \theta_e), \tag{1}$$

where $L(\theta_l, \theta_e) \triangleq -\sum_{\zeta_i \in D_{r_{ld}^{\theta_l}, r_e}} \sum_{t=0}^{T-1} [\ln \pi^{\theta_l, \theta_e}(a^{it}|s^{it})] + \frac{\lambda}{2}\|\theta_e\|_2^2, (a^{it}, s^{it}) \in \zeta_i$. This maximum likelihood IRL (ML-IRL) aims to learn a reward function $r_e^{\theta_e^*(\theta_l)}$ that makes the sample demonstrations from $D_{r_{ld}^{\theta_l}, r_e}$ most probable. Notice that the maximum likelihood estimation has been widely used in IRL to infer the reward function of the expert (Ziebart et al., 2008; 2010). As we prefer a simpler reward function, a $L_2$ regularization term $\frac{\lambda}{2}\|\theta_e\|_2^2$ is added. From the optimization problem (1), a reward parameter $\theta_e^*(\theta_l)$ will be learned. With $r_{ld}^{\theta_l}$, the policy $\pi^{\theta_l, \theta_e^*(\theta_l)}$ corresponding to the learned expert reward function $r_e^{\theta_e^*(\theta_l)}$ is expected to produce the same trajectory from the policy $\pi^{r_{ld}^{\theta_l}, r_e}$ corresponding to the ground truth expert reward function $r_e$.

**Learning expert-dependent reward function $r_{ld}$.** Given $\theta_l$, the learner solves problem (1) to obtain $\theta_e^*(\theta_l)$. So $\theta_e^*$ can be viewed as a best response mapping. For the time being, let us assume that the learner knows the mapping $\theta_e^*$. The learner aims to choose parameter $\theta_l$ that maximizes its cumulative reward $E^{\pi^{\theta_l, \theta_e^*(\theta_l)}}[\sum_{t=0}^{T-1} \gamma^t(r_{ld}(s^t, a^t) + r_{li}(s_l^t, a_l^t))]$. Notice that the optimal policies $\pi^{\theta_l, \theta_e^*(\theta_l)}$ of the MG $(S, A, P, T, r_{ld}^{\theta_l} + r_{li}, r_e^{\theta_e^*(\theta_l)}, \gamma)$ only depend on $\theta_l$. So finding $\theta_l$ can be viewed as an RL problem where the policy is parameterized by $\theta_l$. The learning process is shown in the lower block of Figure 1 and formulated as follows:

$$\max_{\theta_l \in \Theta_l} \quad f(\theta_l, \theta_e^*(\theta_l)) \triangleq E^{\pi^{\theta_l, \theta_e^*(\theta_l)}}\Big[\sum_{t=0}^{T-1} \gamma^t(r_{ld}(s^t, a^t) + r_{li}(s_l^t, a_l^t))\Big] \tag{2}$$

Recall that the learner is unaware of the reward function $r_{ld}$ but can access the value of $r_{ld}(s, a)$ given any state-action pair $(s, a)$. The parameter $\theta_l$ is learned in the same way as the policy gradient by calculating $\nabla_{\theta_l} f(\theta_l, \theta_e^*(\theta_l))$ since the policy is parameterized by $\theta_l$.

**Overall learning process.** The overall learning process is a hierarchical structure, the learner has the knowledge that the expert interacts with the best response $\theta_e^*(\theta_l)$ from problem (1). The learner learns $\theta_l^*$ from problem (2) where the learner is able to predict the response $\theta_e^*(\theta_l)$. This hierarchical structure is shown in Figure 1 and is formulated as the following bi-level optimization problem where the lower-level learns the expert reward function $r_e^{\theta_e^*(\theta_l)}$ which corresponds to the expert-dependent reward function $r_{ld}^{\theta_l}$ from the upper-level and the upper-level learns the expert-dependent reward function $r_{ld}^{\theta_l^*}$.

$$\max_{\theta_l \in \Theta_l} \quad f(\theta_l, \theta_e^*(\theta_l)), \quad \text{s.t.} \quad \theta_e^*(\theta_l) = \arg\min_{\theta_e \in \Theta_e} \quad L(\theta_l, \theta_e). \tag{3}$$

## 6 Algorithm

In this section, we develop a double-loop algorithm GSIIRL, Algorithm 1, to solve the bi-level optimization problem (3). In each iteration, the learner partially solves the lower-level optimization problem through the inner loop (lines 3-8) and then uses the sub-optimal lower-level solution to solve the upper-level optimization problem through the outer loop (lines 9-15). Following this order, we present the details of the inner loop in Section 6.1 and those of the outer loop in Section 6.2.

---

**Algorithm 1** General Sum Interactive Inverse Reinforcement Learning (GSIIRL)

---

1: Initializes $\theta_l(0) \in \theta_l$, $\theta_e(0) \in \theta_e$, step size sequence $\{\alpha_k\}_{k \geq 0}$, $\{\beta_t\}_{t \geq 0}$, regularization parameter $\lambda$ and integer sequence $\{t_k\}_{k \geq 0}$
2: **for** $k = 0, 1, \cdots, K - 1$ **do**
3: $\quad \theta_e(0) = \theta_e(k)$
4: $\quad$ Samples the trajectory set $D_{r_{ld}^{\theta_l(k)}, r_e}$
5: $\quad$ **for** $t = 0, \cdots, t_k - 1$ **do**
6: $\quad\quad$ Samples trajectories under the policy $\pi^{\theta_l(k), \theta_e(t)}$
7: $\quad\quad \theta_e(t+1) = \arg\min_{u \in \theta_e} \langle \nabla_{\theta_e} L(\theta_l(k), \theta_e(t)), u \rangle + \frac{1}{2\beta_t} \|u - \theta_e(t)\|^2$
8: $\quad$ **end for**
9: $\quad \theta_e(k) = \theta_e(t_k - 1)$
10: $\quad$ Samples trajectories under the policy $\pi^{\theta_l(k), \theta_e(k)}$
11: $\quad$ Initializes the random vector $\Delta(k)$ and the positive scalar $p(k)$
12: $\quad$ Gets $\hat{\nabla}^2_{\theta_l \theta_e} L(\theta_l(k), \theta_e(k))$, $\hat{\nabla}^2_{\theta_e} L(\theta_l(k), \theta_e(k))$, $\hat{\nabla}_{\theta_l} f(\theta_l, \theta_e)$ and $\hat{\nabla}_{\theta_e} f(\theta_l, \theta_e)$ through SPSA approximation following equation (4)
13: $\quad [\hat{\nabla}^2_{\theta_e \theta_e} L(\theta_l(k), \theta_e(k))]^{-1} \hat{\nabla}_{\theta_e} f(\theta_l(k), \theta_e(k)) = \min_u \frac{1}{2} u^T \hat{\nabla}^2_{\theta_e} L(\theta_l(k), \theta_e(k)) u$
$\quad\quad - u^T \hat{\nabla}_{\theta_e} f(\theta_l, \theta_e)$
14: $\quad \hat{\nabla} f(\theta_l(k), \theta_e(k)) = \hat{\nabla}_{\theta_l} f(\theta_l(k), \theta_e(k))$
$\quad\quad - \hat{\nabla}^2_{\theta_l \theta_e} L(\theta_l(k), \theta_e(k)) [\hat{\nabla}^2_{\theta_e} L(\theta_l(k), \theta_e(k))]^{-1} \hat{\nabla}_{\theta_e} f(\theta_l(k), \theta_e(k))$
15: $\quad \theta_l(k+1) = \arg\min_{u \in \theta_l} \langle \hat{\nabla} f(\theta_l(k), \theta_e(k)), u \rangle + \frac{1}{2\alpha_k} \|u - \theta_l(k)\|^2$
16: **end for**

---

### 6.1 Inner loop

At each inner loop iteration $t$, the learner uses the projected SGD to update $\theta_e(t)$, i.e., $\theta_e(t)$ moves along the opposite direction of the partial gradient $\nabla_{\theta_e} L(\theta_l(k), \theta_e(t))$ and is then projected onto the set $\theta_e$. The following lemma shows the analytical expression of $\nabla_{\theta_e} L(\theta_l, \theta_e)$ and its proof is given in the appendix. The computation of $\nabla_{\theta_e} L(\theta_l, \theta_e)$ requires the expected reward gradient of the expert $\mu_e(\pi^{\theta_l, \theta_e}) \triangleq E^{\pi^{\theta_l, \theta_e}} [\sum_{t=0}^{T-1} \gamma^t \nabla_{\theta_e} r_e^{\theta_e}(s^t, a^t)]$ and its empirical estimate $\hat{\mu}_e(D_{r_{ld}^{\theta_l}, r_e}) \triangleq \frac{1}{d} \sum_{i=0}^{d} \sum_{t=0}^{T-1} \gamma^t \nabla_{\theta_e} r_e^{\theta_e}(s^{it}, a^{it}), (s^{it}, a^{it}) \in \zeta_i$.

**Lemma 1.** *The gradient* $\nabla_{\theta_e} L(\theta_l, \theta_e) = \mu_e(\pi^{\theta_l, \theta_e}) - \hat{\mu}_e(D_{r_{ld}^{\theta_l}, r_e}) + \lambda \theta_e$

Among $t_k$ iterations, the expectation $\hat{\mu}_e(D_{r_{ld}^{\theta_l(k)}, r_e})$ remains unchanged but $\mu_e(\pi^{\theta_l(k), \theta_e(t)})$ needs to be recalculated for each $\theta_e(t)$. The inner loop terminates after $t_k$ iterations and the estimate $r_e^{\theta_e(t_k-1)}$ is used for the update of the outer loop at the $k$-th iteration.

### 6.2 Outer loop

At the $k$-th iteration of the outer loop, the learner aims to update $\theta_l(k)$ through the projected SGD. Analogous to the inner loop, the projected SGD requires to compute the hypergradient $\nabla f(\theta_l(k), \theta_e(k))$. The analytical expression of $\nabla f(\theta_l(k), \theta_e(k))$ is shown in the following lemma and it is widely used in the bi-level optimization problems. The computation of $\nabla f(\theta_l(k), \theta_e(k))$

requires the partial gradients $\nabla_{\theta_l} f(\theta_l, \theta_e)$, $\nabla_{\theta_e} f(\theta_l, \theta_e)$, the Jacobian $\nabla^2_{\theta_l \theta_e} L(\theta_l, \theta_e)$ and the inverse Hessian $[\nabla^2_{\theta_e} L(\theta_l, \theta_e)]^{-1}$. However, the large computational complexity, $\mathcal{O}(T^3)$, of directly computing each is a challenge to the learner. To solve this challenge, we approximate them through SPSA and reduce computational complexities to $\mathcal{O}(T^2)$. The detail for computational complexities decreasing is discussed later in the Theorem 1. Then the learner uses $\hat{\nabla} f(\theta_l(k), \theta_e(k))$ for the projected gradient.

**Lemma 2.** *The hypergradient of $f(\theta_l, \theta_e)$ for updating $\theta_l$ is $\nabla_{\theta_l} f(\theta_l, \theta_e) - \nabla^2_{\theta_l \theta_e} L(\theta_l, \theta_e)[\nabla^2_{\theta_e} L(\theta_l, \theta_e)]^{-1} \nabla_{\theta_e} f(\theta_l, \theta_e)$*

Take $\hat{\nabla}^2_{\theta_e} L(\theta_l(k), \theta_e(k))$ as an example to illustrate SPSA. According to the equation (2.2) in (Spall, 1992), it is approximated as follows:

$$\hat{\nabla}^2_{\theta_e} L(\theta_l(k), \theta_e(k)) = \begin{bmatrix} \frac{\nabla_{\theta_e} L(\theta_l(k), \theta_e(k) + p(k)\Delta(k)) - \nabla_{\theta_e} L(\theta_l(k), \theta_e(k) - p(k)\Delta(k))}{2p\Delta_1(k)} \\ \vdots \\ \frac{\nabla_{\theta_e} L(\theta_l(k), \theta_e(k) + p(k)\Delta(k)) - \nabla_{\theta_e} L(\theta_l(k), \theta_e(k) - p(k)\Delta(k))}{2p\Delta_m(k)} \end{bmatrix}, \quad (4)$$

the perturbation $\Delta(k) \in \mathbb{R}^m$ is a vector of $m$ mutually independent zero-mean random variables and each element of $\Delta(k)$ satisfies $|\Delta_i(k)| \leq \alpha_0$, $E|\Delta_i^{-1}(k)| \leq \alpha_l$, $i = 1, \cdots, m$ with $\alpha_0, \alpha_l$ as positive constants. The parameter $p(k)$ is a positive scalar. The computation of $\hat{\nabla}_{\theta_l} f(\theta_l(k), \theta_e(k))$ requires $f(\theta_l(k), \theta_e(k))$ with weight perturbations on $\theta_l(k)$. Analogously, computing $\hat{\nabla}_{\theta_e} f(\theta_l(k), \theta_e(k))$, $\hat{\nabla}^2_{\theta_e} L(\theta_l(k), \theta_e(k))$, and $\hat{\nabla}^2_{\theta_l \theta_e} L(\theta_l(k), \theta_e(k))$ requires $f(\theta_l(k), \theta_e(k))$, $\nabla_{\theta_e} L(\theta_l(k), \theta_e(k))$, and $\nabla_{\theta_l} L(\theta_l(k), \theta_e(k))$ with weight perturbations on $\theta_e(k)$, respectively. The analytical expression of $f(\theta_l(k), \theta_e(k))$ and $\nabla_{\theta_e} L(\theta_l(k), \theta_e(k))$ are shown in optimization problem (2) and Lemma 1 respectively. Similarly to Lemma 1, the gradient $\nabla_{\theta_l} L(\theta_l(k), \theta_e(k)) \triangleq \mu_l(\pi^{\theta_l(k), \theta_e(k)}) - \hat{\mu}_l(D_{r_{ld}^{\theta_l(k)}, r_e})$ and the proof is given in the appendix. The expectation $\mu_l(\pi^{\theta_l, \theta_e}) \triangleq E^{\pi^{\theta_l, \theta_e}}[\sum_{t=0}^{T-1} \gamma^t \nabla_{\theta_l} r_{ld}^{\theta_l}(s^t, a^t)]$ is the reward gradient expectation of the learner and the expectation $\hat{\mu}_l(D_{r_{ld}^{\theta_l}, r_e}) \triangleq \frac{1}{d} \sum_{i=0}^{d} \sum_{t=0}^{T-1} \gamma^t \nabla_{\theta_l} r_{ld}^{\theta_l}(s^{it}, a^{it})$, $(s^{it}, a^{it}) \in \zeta_i$ is the estimated reward gradient expectation of the learner.

In order to successfully to compute $\hat{\nabla} f(\theta_l(k), \theta_e(k))$, the existence of inverse Hessian $[\hat{\nabla}^2_{\theta_e} L(\theta_l(k), \theta_e(k))]^{-1}$ needs to be guaranteed. Since the negative log-likelihood function is convex and the $L_2$ regularization term is obviously strongly convex, the overall function $L(\theta_l(k), \theta_e(k))$ is strongly convex. We can conclude that $\nabla^2_{\theta_e} L(\theta_l(k), \theta_e(k)) \geq \lambda I$. Then, we can choose proper $\Delta(k)$ and $p(k)$ which lead to $\hat{\nabla}^2_{\theta_e} L(\theta_l(k), \theta_e(k)) > 0$. As directly computing the inverse of a matrix is computational, we use the conjugate gradient method to compute $[\hat{\nabla}^2_{\theta_e \theta_e} L(\theta_l(k), \theta_e(k))]^{-1} \hat{\nabla}_{\theta_e} f(\theta_l(k), \theta_e(k))$ and the expression is shown in line 13. With $\hat{\nabla}_{\theta_l} f(\theta_l(k), \theta_e(k))$, $\hat{\nabla}^2_{\theta_l \theta_e} L(\theta_l(k), \theta_e(k))$ and $[\hat{\nabla}^2_{\theta_e \theta_e} L(\theta_l(k), \theta_e(k))]^{-1} \hat{\nabla}_{\theta_e} f(\theta_l(k), \theta_e(k))$, the estimate hypergradient $\hat{\nabla} f(\theta_l(k), \theta_e(k)) = \hat{\nabla}_{\theta_l} f(\theta_l(k), \theta_e(k)) - \hat{\nabla}^2_{\theta_l \theta_e} L(\theta_l(k), \theta_e(k))[\hat{\nabla}^2_{\theta_e} L(\theta_l(k), \theta_e(k))]^{-1} \hat{\nabla}_{\theta_e} f(\theta_l(k), \theta_e(k))$ can be calculated.

After $K$ iterations, the outer loop terminates and the GSIIRL outputs $r_{ld}^{\theta_l(K)}$, $r_e^{\theta_e(K)}$ and $\pi^{\theta_l(K), \theta_e(K)}$.

# 7 ANALYTICAL RESULT

In this section, we conclude the analytical results for computational complexity reduction and the convergence rate.

We add an assumption on the estimated reward functions $r_{ld}^{\theta_l}$ and $r_e^{\theta_e}$ for the analytical results. The estimated reward functions satisfy the following assumption.

**Assumption 1.** *The estimated expert-dependent reward function $r_{ld}^{\theta_l}$, and the estimated expert reward function $r_e^{\theta_e}$ are fourth differentiable, i.e., $C^4$.*

Since $\theta_l$ and $\theta_e$ are compact, Assumption 1 implies the first-order to the fourth-order gradients of the reward functions $r_{ld}^{\theta_l}$ and $r_e^{\theta_e}$ are bounded. The boundedness assumption is widely adopted in

bi-level optimization (Jin et al., 2020; Zeng et al., 2022; Liu & Zhu, 2023a), RL (Wang et al., 2019; Zhang et al., 2020) and IRL (Liu & Zhu, 2023b).

## 7.1 COMPUTATIONAL COMPLEXITY

The computation complexities of computing $\nabla f(\theta_l, \theta_e)$ and $\hat{\nabla} f(\theta_l, \theta_e)$ are quantified in Theorem 1

**Theorem 1.** *Consider $T$ as the decision factor, the computational complexity of computing $\nabla f(\theta_l, \theta_e)$ is $\mathcal{O}(T^3)$ and that of computing $\hat{\nabla} f(\theta_l, \theta_e)$ is $\mathcal{O}(T^2)$.*

By applying the SPSA to approximate $\nabla f(\theta_l, \theta_e)$, the computational complexity for each iteration of the upper-level reduces from $\mathcal{O}(T^3)$ to $\mathcal{O}(T^2)$. The proof of Theorem 1 is in the appendix. Compared to the finite difference, SPSA requires fewer policies for the approximation. In SPSA, a random perturbation is added or subtracted to $\theta_l$ or $\theta_e$, and therefore we only need two policies for the approximation. In finite difference, two policies are required for each dimension of $\theta_l$ or $\theta_e$. These policies are generated through RL, so the computational cost of generating the policies can not be neglected. As SPSA requires fewer policies, the computational cost of SPSA is lower than the finite difference which has a lower computational cost than directly computing gradients.

## 7.2 CONVERGENCE RATE

This section shows the main result of our algorithm, the converge result is shown in Theorem 2.

**Theorem 2.** *Suppose Assumption 1 holds, by choosing $p(k) = \frac{1}{k}$, $\alpha_k = \frac{1}{L_f \sqrt{K}}$, $t_k = \lceil \frac{\sqrt[4]{k+1}}{2} \rceil$, the following result holds: $\frac{1}{K} \sum_{k=0}^{K-1} E[\|\nabla f(\theta_l(k), \theta_e^*(\theta_l(k)))\|^2] \leq \mathcal{O}(\frac{1}{\sqrt{K}})$.*

**Corollary 2.1.** *If the reward function $r_e^{\theta_e}$ are linear, the cumulative reward difference between the learned expert policy $\pi_{\theta_e}$ and $\pi_e$ decreases at a rate $\mathcal{O}(\frac{1}{\sqrt[4]{K}})$.*

The proof of Theorem 2 and Corollary 2.1 are shown in the appendix. The Theorem 2 indicates that the expected hypergradient decreases at the rate of $\mathcal{O}(\frac{1}{\sqrt{K}})$. It is the same rate as the projected SGD, so the bias from the SPSA does not slow the convergence rate compared to standard projected SGD. The Corollary 2.1 indicates the convergence of the cumulative reward difference between $\pi_{\theta_e}$ and $\pi_e$ which has been widely used to infer the convergence of the learned expert policy.

## 8 EXPERIMENT

This section conducts three experiments to evaluate the performance of the GSIIRL. We compare the GSIIRL with two benchmarks: the **MARL** algorithm (Lowe et al., 2017) and the **MA-IRL** algorithm (Lin et al., 2019). The MARL uses the ground truth reward functions of the learner and the expert and the other two algorithms use the learned reward functions. To learn the reward functions, the MA-IRL requires demonstrations generated through the ground truth reward functions of the learner and the expert. The GSIIRL only requires the interactions of the expert and the accessed values of the expert-dependent reward function. Recall that the learner aims to learn the reward function of the expert from observed trajectories and an expert-dependent reward function from accessed values.

## 8.1 MULTI-AGENT PARTICLE ENVIRONMENTS

We use the physical deception environment which is a non-cooperative environment in the Multi-Agent Particle Environments (MPE) (Lowe et al., 2017; Terry et al., 2021) to do the simulation. There are one adversary, two cooperating agents, and two landmarks. One of the landmarks is the target landmark, the cooperating agents know which one is the target landmark but the adversary does not know. The cooperating agents aim to let one of them get as close as the target landmark and keep the adversary as far as the target landmark. Therefore, the cooperating agents need to learn how to spread out and cover both landmarks to deceive the adversary. The cooperating agents are viewed as a single learner and the adversary is the expert. The accessed values are the distances between the adversary and the target landmark. The state space dimensions for the agents and the

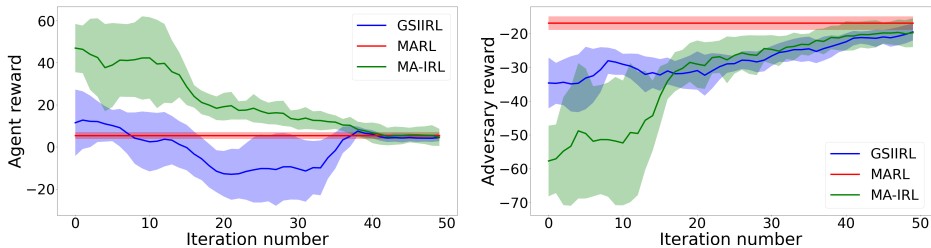

Figure 2: MPE simulation results. **Left**: Agent's reward. **Right**: Adversary's reward. The horizontal lines are the results from the MARL with ground truth reward functions. For the MA-IRL and the GSIIRL, the reward functions are updated at each iteration. In each iteration, we sample the trajectories of the adversary and the agents through MARL with updated reward functions and plot the cumulative rewards of the sampled trajectories with ground truth reward functions.

adversary are $10$ and $8$ respectively, and the action space dimensions for the agents and the adversary are $5$. Other experiment details are listed in Appendix Section A.10.1.

We compare the performance of the algorithms through the cumulative rewards. Since two agents cooperate with each other and share the same reward, we only plot one graph for the agents' cumulative reward. In Figure 2, two randomly initialized reward functions are updated during 50 iterations. At the beginning, the adversary and the agents move randomly and the initial cumulative rewards are far from the horizontal line. While updating reward functions, the agents learn the goals of themselves and the adversary, then the cumulative rewards converge to the horizontal line.

## 8.2 HUMAN-ROBOT INTERACTION

Consider the human-robot interaction scenario in Section 3, the robot (learner) and the human (expert) aim to reach their destinations as soon as possible. In addition, the robot needs to keep the human with positive emotions. The state space dimensions and the action space dimensions for the robot and the human are $4$ and $2$ respectively. The accessed values are the distance between the robot and the human and the human's speed, detailed setups are shown in Appendix Section A.10.2.

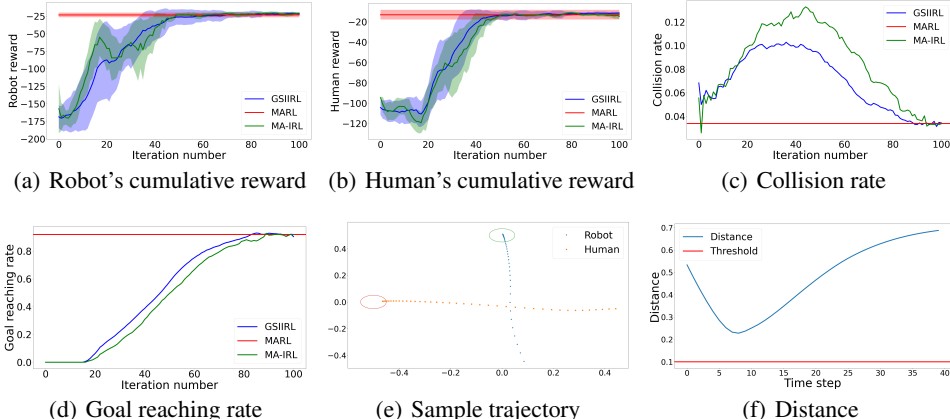

Figure 3: Human-robot interaction simulation results. The cumulative rewards, the collision rate, and the goal reaching rate are calculated in the same way described in Figure 2.

We use four metrics to compare the algorithms: the robot's cumulative reward, the human's cumulative reward, the collision rate, and the goal-reaching rate. Then we show a sample trajectory from the results of the GSIIRL. In Figure 3(a) and Figure 3(b), the cumulative rewards of the robot and the human converge to the ground truth values which means the robot reaches its maximum cumulative reward and successfully learns the human behaviors. Figure 3(c) shows the collision rate coverages to the ground truth value which means the robot successfully learns a human-dependent reward function. In the beginning, the reward functions are randomly generated. The robot and human move randomly and the collision rarely happens. While updating the reward functions, the robot and the human move toward the goal locations, and collisions occur which leads to an increas-

ing collision rate. Then the reward functions are better learned to avoid collision and the collision rate decreases. From Figure 3(d), we can see that the goal-reaching rate converges to the ground truth value. With the learned reward functions, both the robot and the human can reach their goals. Overall, the learned robot's reward function leads to a high goal-reaching probability with a low collision probability after 100 iteration updates. Figure 3(e) shows a sample trajectory with learned reward functions and Figure 3(f) plots the distances between the robot and the human in Figure 3(e). The robot and the human reach their goals and keep the distance above the safe distance.

## 8.3 SECURITY

We run a cyber-security experiment to test the proposed algorithm. Consider a defender (learner) and an attacker (expert) interacting in an attack graph in Figure 4. Detailed setups are shown in the Appendix Section A.10.3. The defender aims to protect the system and the attacker aims to attack the system as much as possible. The accessed value for the defender is the negative value of the attacker's reward. The cardinalities for the discrete state space and the discrete action space are 256 and 8 respectively.

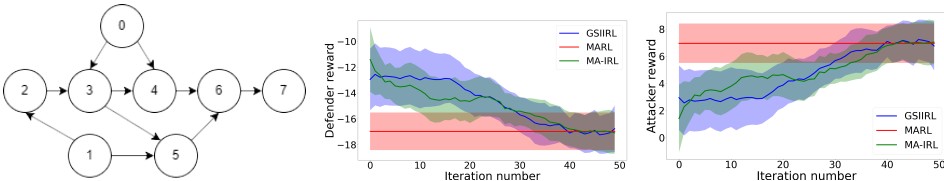

Figure 4: Cyber security simulation results. **Left**: Attack graph. **Middle**: Defender's reward. **Right**: Attacker's reward. The cumulative rewards are calculated in the same way described in Figure 2.

From Figure 4, the randomly generated reward functions are updated through 50 iterations. Starting with random reward functions, both the defender and the attacker act randomly. Therefore, the cumulative reward of the defender is high and that of the attacker is low at the beginning. As the reward functions keep being updated, the attacker starts to attack more nodes, and the defender tries to block the attacks. Finally, the cumulative rewards converge to the ground truth values.

## 8.4 RESULTS ANALYSIS

In this section, we summarize the final results of the cumulative rewards in three experiments as follows:

| Learner | MPE | HRI | CS |
|---|---|---|---|
| MARL | $5.43 \pm 1.66$ | $-20.22 \pm 3.34$ | $-16.96 \pm 1.45$ |
| MA-IRL | $5.12 \pm 4.08$ | $-20.63 \pm 1.07$ | $-17.08 \pm 3.21$ |
| GSIIRL | $4.54 \pm 1.81$ | $-21.69 \pm 1.87$ | $-17.38 \pm 3.59$ |
| Expert | MPE | HRI | CS |
| MARL | $-16.91 \pm 2.01$ | $-12.98 \pm 4.78$ | $6.96 \pm 1.45$ |
| MA-IRL | $-20.37 \pm 4.27$ | $-13.52 \pm 2.57$ | $7.03 \pm 3.21$ |
| GSIIRL | $-19.53 \pm 2.50$ | $-13.16 \pm 1.03$ | $6.76 \pm 3.59$ |

Table 1: The cumulative reward comparison. **Top**: The cumulative reward of the learner. **Bottom**: The cumulative reward of the expert. MARL uses ground truth reward functions. MA-IRL and GSIIRL use learned reward functions from the last iteration.

From Table 1 and Figure (2), (3), (4), the cumulative reward results of the MA-IRL and the GSIIRL coverage to the results of the MARL. We can see the cumulative reward results of the GSIIRL is comparable to that of the MA-IRL while the GSIIRL requires less information than the MA-IRL. The GSIIRL only requires the interactions of the expert and the accessed values of the expert-dependent reward function instead of the demonstrations generated through the ground truth reward functions of the learner and the expert for the MA-IRL.

## 9 CONCLUSION

This paper develops a general-sum interactive inverse reinforcement learning framework to learn the reward functions and policies of the learner and the expert during the interactions. This framework includes the fully cooperative framework and the fully competitive framework as special cases. Then we propose the GSIIRL algorithm and theoretically quantify its convergence rate. From the experiments, we show the effectiveness of the GSIIRL in both continuous and discrete environments.

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

# A APPENDIX

## A.1 NOTION AND NOTATIONS

Define $f(\theta_l, \theta_e) \triangleq J_{ed}(\pi^{\theta_l,\theta_e}) + J_{ei}(\pi^{\theta_l,\theta_e})$, where $J_{ed}(\pi^{\theta_l,\theta_e}) \triangleq E^{\pi^{\theta_l,\theta_e}}[\sum_{t=0}^{T-1} \gamma^t r_{ld}^{\theta_l}(s^t, a^t)]$ is the cumulative expert-dependent reward of the learner and $J_{ei}(\pi^{\theta_l,\theta_e}) \triangleq$ $E^{\pi^{\theta_l,\theta_e}}[\sum_{t=0}^{T-1} \gamma^t r_{li}(s_l^t, a_l^t)]$ is the cumulative expert-independent reward of the learner. Define the reward gradient expectation of the expert from the state-action pair $(s, a)$ as $\mu_e(s, a) \triangleq E^{\pi^{\theta_l,\theta_e}}[\sum_{t=0}^{T-1} \gamma^t \nabla_{\theta_e} r_e^{\theta_e}(s^t, a^t)|s^0 = s, a^0 = a]$, the reward gradient expectation of the expert from the state $s$ as $\mu_e(s) \triangleq \int_{a_l \in a_l} \int_{a_e \in a_e} \mu_e(s, a) da_l da_e$, $\mu_e(s, a) \triangleq E^{\pi^{\theta_l,\theta_e}}[\sum_{t=0}^{T-1} \gamma^t \nabla_{\theta_e} r_e^{\theta_e}(s^t, a^t)|s^0 = s]$. Analogously, define the reward gradient expectation of the learner as $\mu_l(\pi^{\theta_l,\theta_e}) \triangleq E^{\pi^{\theta_l,\theta_e}}[\sum_{t=0}^{T-1} \gamma^t \nabla_{\theta_l} r_{ld}^{\theta_l}(s^t, a^t)]$, the reward gradient expectation of the learner from the state-action pair $(s, a)$ as $\mu_l(s) \triangleq E^{\pi^{\theta_l,\theta_e}}[\sum_{t=0}^{T-1} \gamma^t \nabla_{\theta_l} r_{ld}^{\theta_l}(s^t, a^t)|s^0 = s]$, $\mu_l(s, a) \triangleq E^{\pi^{\theta_l,\theta_e}}[\sum_{t=0}^{T-1} \gamma^t \nabla_{\theta_l} r_{ld}^{\theta_l}(s^t, a^t)|s^0 = s, a^0 = a]$, the reward gradient expectation of the learner from the state $s$ as $\mu_l(s) \triangleq \int_{a_l \in a_l} \int_{a_e \in a_e} \mu_l(s, a) da_l da_e$. Define the cumulative expert-independent reward of the learner from the state-action pair $(s, a)$ as $J_{ei}(s, a) \triangleq E^{\pi^{\theta_l,\theta_e}}[\sum_{t=0}^{T-1} \gamma^t r_{li}(s_l^t, a_l^t)|s^0 = s, a^0 = a]$ and define the cumulative expert-independent reward of the learner from the state $s$ as $J_{ei}(s) \triangleq E^{\pi^{\theta_l,\theta_e}}[\sum_{t=0}^{T-1} \gamma^t r_{li}(s_l^t, a_l^t)|s^0 = s]$. Define $P_0(s)$ as the probability distribution of the initial state.

During the proofs, symbols $(i)$ to $(viii)$ are used to represent what theorems or methods are used to get the current step. The symbol $(i)$ represents chain rule, the symbol $(ii)$ represents the linearity of expectation, the symbol $(iii)$ represents the close form of geometric series, the symbol $(iv)$ represents the triangle inequality, the symbol $(v)$ represents the hölder's inequality, the symbol $(vi)$ represents Taylor's theorem, the symbol $(vii)$ means the usage of other equations in this paper, and the symbol $(viii)$ means keeping expansion.

## A.2 FUNDAMENTAL RESULT FOR POLICY

This section lists the expressions for important gradients in continuous state-action space. Based on the idea of the soft Q learning (Haarnoja et al., 2017), we can get:

$$Q_{\theta_l,\theta_e}^{soft}(s, a) = r_{ld}^{\theta_l}(s, a) + r_{li}(s, a) + r_e^{\theta_e}(s, a) + \gamma \int_{s' \in S} P(s'|s, a) V^{soft}(s') ds', \quad (5)$$

$$V_{\theta_l,\theta_e}^{soft}(s) = \ln \int_{a_l \in a_l} \int_{a_e \in a_e} \exp(Q^{soft}(s, a)) da_l da_e, \quad (6)$$

$$\pi^{\theta_l,\theta_e}(a_l, a_e|s) = \frac{\exp(Q^{soft}(s, a))}{\exp(V^{soft}(s))}. \quad (7)$$

As the lower-level optimization problem is a maximum likelihood problem, it is important to know $\nabla_{\theta_e} \ln(\pi^{\theta_l,\theta_e})$ for the SGD of the lower-level optimization problem. The process for getting $\nabla_{\theta_e} \ln(\pi^{\theta_l,\theta_e})$ is shown below.
From the expression of $\pi^{\theta_l,\theta_e}$, we can get $\nabla_{\theta_e} \ln(\pi^{\theta_l,\theta_e}) = \nabla_{\theta_e} Q^{soft}(s, a) - \nabla_{\theta_e} V^{soft}(s)$, therefore, we can calculate $\nabla_{\theta_e} Q^{soft}(s, a)$ and $\nabla_{\theta_e} V^{soft}(s)$ separately.

Based on the equation of $V^{soft}(s)$, the gradient $\nabla_{\theta_e} V^{soft}(s)$ could be calculated as follows:

$$
\begin{aligned}
&\nabla_{\theta_e} V^{soft}(s), \\
&\overset{(i)}{=} \frac{\int_{a_l \in a_l} \int_{a_e \in a_e} \nabla_{\theta_e} \exp(Q^{soft}(s,a)) da_l da_e}{\int_{a_l \in a_l} \int_{a_e \in a_e} \exp(Q^{soft}(s,a)) da_l da_e}, \\
&\overset{(vii)}{=} \frac{\int_{a_l \in a_l} \int_{a_e \in a_e} \exp(Q^{soft}(s,a)) \nabla_{\theta_e} Q^{soft}(s,a) da_l da_e}{\exp(V^{soft}(s))}, \\
&\overset{(vii)}{=} \int_{a_l \in a_l} \int_{a_e \in a_e} \pi^{\theta_l,\theta_e}(a_l, a_e|s)(\nabla_{\theta_e} r_e(s,a) \\
&\quad + \gamma \int_{s' \in S} P(s'|s,a) \nabla_{\theta_e} V^{soft}(s')ds')da_l da_e, \\
&\overset{(viii)}{=} \int_{a_l \in a_l} \int_{a_e \in a_e} \pi^{\theta_l,\theta_e}(a_l, a_e|s)(\nabla_{\theta_e} r_e(s,a) + \gamma \int_{s' \in S} P(s'|s,a) \\
&\quad (\int_{a_l' \in a_l} \int_{a_e' \in a_e} \pi^{\theta_l,\theta_e}(a_l', a_e'|s')[\nabla_{\theta_e} r_e(s', a_l', a_e') \\
&\quad + \gamma \int_{s'' \in S} P(s''|s', a_l', a_e') \nabla_{\theta_e} V^{soft}(s'')ds'']da_l' da_e' ds')da_l da_e, \\
&= E^{\pi^{\theta_l,\theta_e}} [\sum_{t=0}^{T-1} \gamma^t \nabla_{\theta_e} r_e(s^t, a^t)|s^0 = s], \\
&= \mu_e(s),
\end{aligned}
\tag{8}
$$

where the first $(vii)$ uses the expression in equation (6) and (7), and the second $(iii)$ uses the expression in equation (5).
Similarly, we can get $\nabla_{\theta_e} Q^{soft}(s,a)$ from $Q^{soft}(s,a)$.

$$
\begin{aligned}
&\nabla_{\theta_e} Q^{soft}(s,a), \\
&\overset{(vii)}{=} \nabla_{\theta_e} r_e(s,a) + \gamma \int_{s' \in S} P(s'|s,a) \nabla_{\theta_e} V^{soft}(s')ds')da_l da_e, \\
&\overset{(viii)}{=} E^{\pi^{\theta_l,\theta_e}} [\sum_{t=0}^{T-1} \gamma^t \nabla_{\theta_e} r_e(s^t, a^t)|s^0 = s, a^0 = a], \\
&= \mu_e(s,a),
\end{aligned}
\tag{9}
$$

where $(vii)$ uses the expression in equation (5).
By summing the results of equation (8) and (9), we can get $\nabla_{\theta_e} \ln(\pi^{\theta_l,\theta_e})$ as follows:

$$
\nabla_{\theta_e} \ln(\pi^{\theta_l,\theta_e}(a_l, a_e|s)) = \nabla_{\theta_e} Q^{soft}(s,a) - \nabla_{\theta_e} V^{soft}(s) = \mu_e(s,a) - \mu_e(s).
\tag{10}
$$

The way to get $\nabla_{\theta_l} \ln(\pi^{\theta_l,\theta_e})$ is same as that of $\nabla_{\theta_e} \ln(\pi^{\theta_l,\theta_e})$, and the gradient $\nabla_{\theta_l} \ln(\pi^{\theta_l,\theta_e}(a_l, a_e|s)) = \mu_l(s,a) - \mu_l(s)$.

## A.3 THE PROOF OF LEMMA 1

In this section, we derived gradients that are necessary for our method.

Define $P_D(s^t = s, a_l^t = a_l, a_e^t = a_e) \triangleq \begin{cases} 1 & s^t = s, a_l^t = a_l, a_e^t = a_e \\ 0 & \text{otherwise} \end{cases}$, where $(s,a) \in D_{r_{ld}^{\theta_l},r_e}$, as the probability of $(s,a)$ occurring at time $t$ in the demonstration set $D_{r_{ld}^{\theta_l},r_e}$. With the fundamental

result of the policy, we can derive the $\nabla_{\theta_e} L(\theta_l, \theta_e)$ and prove the Lemma 1 as follows:

$$\nabla_{\theta_e} L(\theta_l, \theta_e),$$

$$= -\sum_{t=0}^{T-1} \gamma^t \int_{s \in S} \int_{a_l \in a_l} \int_{a_e \in a_e} P_D(s^t = s, a_l^t = a_l, a_e^t = a_e) \nabla_{\theta_e} \ln(\pi^{\theta_l, \theta_e}) da_e da_l ds + \lambda \theta_e,$$

$$\overset{(vii)}{=} -\sum_{t=0}^{T-1} \gamma^t \int_{s \in S} \int_{a_l \in a_l} \int_{a_e \in a_e} P_D(s^t = s, a_l^t = a_l, a_e^t = a_e)(\nabla_{\theta_e} r_e(s, a)$$

$$+ E^{\pi^{\theta_l, \theta_e}}[\sum_{t'=t+1}^{T-1} \gamma^{t'-t-1} \nabla_{\theta_e} r_e(s^t, a^t) | s^t = s, a_l^t = a_l, a_e^t = a_e]$$

$$- E^{\pi^{\theta_l, \theta_e}}[\sum_{t'=t}^{T-1} \gamma^{t'-t} \nabla_{\theta_e} r_e(s^t, a^t) | s^t = s]) da_e da_l ds + \lambda \theta_e,$$

$$\overset{(ii)}{=} -\sum_{t=0}^{T-1} \gamma^t \int_{s \in S} \int_{a_l \in a_l} \int_{a_e \in a_e} P_D(s^t = s, a_l^t = a_l, a_e^t = a_e)(\nabla_{\theta_e} r_e(s, a) da_e da_l ds$$

$$- \int_{s \in S} \int_{a_l \in a_l} \int_{a_e \in a_e} P_D(s^0 = s, a_l^0 = a_l, a_e^0 = a_e)$$

$$E^{\pi^{\theta_l, \theta_e}}[\sum_{t=0}^{T-1} \gamma^t \nabla_{\theta_e} r_e(s^t, a^t) | s^0 = s]) da_e da_l ds + \lambda \theta_e,$$

$$= \mu_e(\pi^{\theta_l, \theta_e}) - \hat{\mu}_e(D_{r_{ld}^{\theta_l}, r_e}) + \lambda \theta_e,$$

where $(vii)$ uses the expression of equation (10).

## A.4 Other needed gradients

**Lemma 3.** *Suppose Assumption 1 holds, the first-order gradients $\nabla_{\theta_l} L(\theta_l, \theta_e) = \mu_l(\pi^{\theta_l, \theta_e}) - \hat{\mu}_l(D_{r_{ld}^{\theta_l}, r_e}), \nabla_{\theta_l} f(\theta_l, \theta_e) = E^{\pi^{\theta_l, \theta_e}}[\sum_{t=0}^{T-1} \gamma^t [(\mu_l(s^t, a^t) - \mu_l(s^t))(J_{ed}(s^t, a^t) + J_{ei}(s^t, a^t))^T]] + \mu_l(\pi^{\theta_l, \theta_e}), \nabla_{\theta_e} f(\theta_l, \theta_e) = E^{\pi^{\theta_l, \theta_e}}[\sum_{t=0}^{T-1} \gamma^t [(\mu_e(s^t, a^t) - \mu_e(s^t))(J_{ed}(s^t, a^t) + J_{ei}(s^t, a^t))^T]]. The second-order gradients $\nabla_{\theta_l \theta_e}^2 L(\theta_l, \theta_e) = E^{\pi^{\theta_l, \theta_e}}[\sum_{t=0}^{T-1} \gamma^t (\mu_e(s, a) - \mu_e(s)) \mu_l(s, a)^T], \nabla_{\theta_e \theta_e}^2 L(\theta_l, \theta_e) = E^{\pi^{\theta_l, \theta_e}}[\sum_{t=0}^{T-1} \gamma^t [(\mu_e(s^t, a^t) - \mu_e(s^t)) \mu_e(s^t, a^t)^T + \nabla_{\theta_e}^2 r_e^{\theta_e}(s^t, a^t)]] - \frac{1}{d} \sum_{i=0}^{d} \sum_{t=0}^{T-1} \gamma^t \nabla_{\theta_e} r_e^{\theta_e}(s^{it}, a^{it}) + \lambda$*

*Proof.* Through the same process of calculating $\nabla_{\theta_e} L(\theta_l, \theta_e)$, we can get the gradient $\nabla_{\theta_l} L(\theta_l, \theta_e)$ following the same process in Lemma 1, where $\mu_l(\pi^{\theta_l, \theta_e}) \triangleq E^{\pi^{\theta_l, \theta_e}}[\sum_{t=0}^{T-1} \gamma^t \nabla_{\theta_l} r_{ld}(s^t, a^t)]$.

The process for getting $\nabla_{\theta_l} f(\theta_l, \theta_e)$ is shown below.
From the equation of $f(\theta_l, \theta_e)$, we can see that $\nabla_{\theta_l} f(\theta_l, \theta_e) = \nabla_{\theta_l} J_{ed}(\pi^{\theta_l, \theta_e}) + \nabla_{\theta_l} J_{ei}(\pi^{\theta_l, \theta_e})$, then $\nabla_{\theta_l} J_{ed}(\pi^{\theta_l, \theta_e})$ and $\nabla_{\theta_l} J_{ei}(\pi^{\theta_l, \theta_e})$ can be calculated separately.

The calculation of deriving $\nabla_{\theta_l} J_{ed}(\pi^{\theta_l,\theta_e})$ is as follows:

$$\nabla_{\theta_l} J_{ed}(\pi^{\theta_l,\theta_e}),$$

$$\stackrel{(i)}{=} \int_{s^0 \in S} P_0(s^0) \int_{a_l^0 \in a_l} \int_{a_e^0 \in a_e} [\nabla_{\theta_l} \pi(a_l^0, a_e^0 | s^0) J_{ed}(s^0, a_l^0, a_e^0)^T$$

$$+ \pi(a_l^0, a_e^0 | s^0) \nabla_{\theta_l} J_{ed}(s^0, a_l^0, a_e^0)^T] da_l^0 da_e^0 ds^0,$$

$$= \int_{s^0 \in S} P_0(s^0) \int_{a_l^0 \in a_l} \int_{a_e^0 \in a_e} [\nabla_{\theta_l} \pi(a_l^0, a_e^0 | s^0) J_{ed}(s^0, a_l^0, a_e^0)^T + \pi(a_l^0, a_e^0 | s^0)(\nabla_{\theta_l} r_{ld}^{\theta_l}(s^0, a_l^0, a_e^0)$$

$$+ \gamma \int_{s^1 \in S} P(s^1 | s^0, a_l^0, a_e^0) \nabla_{\theta_l} J_{ed}(s^1)) ds^1] da_l^0 da_e^0 ds^0,$$

$$\stackrel{(viii)}{=} \int_{s^0 \in S} P_0(s^0) \int_{a_l^0 \in a_l} \int_{a_e^0 \in a_e} \{\nabla_{\theta_l} \pi(a_l^0, a_e^0 | s^0) J_{ed}(s^0, a_l^0, a_e^0)^T + \pi(a_l^0, a_e^0 | s^0)[\nabla_{\theta_l} r_{ld}^{\theta_l}(s^0, a_l^0, a_e^0)$$

$$+ \gamma \int_{s^1 \in S} P(s^1 | s^0, a_l^0, a_e^0) \int_{a_l^1 \in a_l} \int_{a_e^1 \in a_e} \nabla_{\theta_l} \pi(a_l^1, a_e^1 | s^1) J_{ed}(s^1, a_l^1, a_e^1)^T$$

$$+ \pi(a_l^1, a_e^1 | s^1)(\nabla_{\theta_l} r_{ld}^{\theta_l}(s^1, a_l^1, a_e^1) + \gamma \int_{s^2 \in S} P(s^2 | s^1, a_l^1, a_e^1) \nabla_{\theta_l} J_{ed}(s^2)) ds^2] ds^1\} da_l^0 da_e^0 ds^0,$$

$$= E^{\pi^{\theta_l,\theta_e}} [\sum_{t=0}^{T-1} \gamma^t (\frac{\nabla_{\theta_l} \pi(a_l^t, a_e^t | s^t)}{\pi(a_l^t, a_e^t | s^t)} J_{ed}(s^t, a^t)^T + \nabla_{\theta_l} r_{ld}^{\theta_l}(s^t, a^t))],$$

$$= E^{\pi^{\theta_l,\theta_e}} [\sum_{t=0}^{T-1} \gamma^t (\nabla_{\theta_l} \ln(\pi^{\theta_l,\theta_e}) J_{ed}(s^t, a^t)^T + \nabla_{\theta_l} r_{ld}^{\theta_l}(s^t, a^t))],$$

$$\stackrel{(vii)}{=} E^{\pi^{\theta_l,\theta_e}} [\sum_{t=0}^{T-1} \gamma^t [(\mu_l(s^t, a^t) - \mu_l(s^t)) J_{ed}(s^t, a^t)^T]] + \mu_l(\pi^{\theta_l,\theta_e}),$$

where $(vii)$ use the expression of the equation (10).
Similarly, we can get $\nabla_{\theta_l} J_{ei}(\pi^{\theta_l,\theta_e})$ as follows:

$$\nabla_{\theta_l} J_{ei}(\pi^{\theta_l,\theta_e}),$$

$$\stackrel{(i)}{=} \int_{s^0 \in S} P_0(s^0) \int_{a_l^0 \in a_l} \int_{a_e^0 \in a_e} [\nabla_{\theta_l} \pi(a_l^0, a_e^0 | s^0) J_{ei}(s^0, a_l^0, a_e^0)$$

$$+ \pi(a_l^0, a_e^0 | s^0)(\nabla_{\theta_l} J_{ei}(s^0, a_l^0, a_e^0))] da_l^0 da_e^0 ds^0,$$

$$\stackrel{(viii)}{=} \int_{s^0 \in S} P_0(s^0) \int_{a_l^0 \in a_l} \int_{a_e^0 \in a_e} [\pi(a_l^0, a_e^0 | s^0) \nabla_{\theta_l} \ln(\pi(a_l^0, a_e^0 | s^0)) J_{ei}(s^0, a_l^0, a_e^0)$$

$$+ \pi(a_l^0, a_e^0 | s^0) \gamma \int_{s^1 \in S} P(s^1 | s^0, a_l^0, a_e^0) \nabla_{\theta_l} J_{ei}(s^1) ds^1] da_l^0 da_e^0 ds^0,$$

$$\stackrel{(vii)}{=} E^{\pi^{\theta_l,\theta_e}} [\sum_{t=0}^{T-1} \gamma^t (\mu_l(s^t, a^t) - \mu_l(s^t)) J_{ei}(s^t, a^t)^T],$$

where $(vii)$ use the expression of the equation (10).
By summing the result of $\nabla_{\theta_l} J_{ed}(\pi^{\theta_l,\theta_e})$ and $\nabla_{\theta_l} J_{ei}(\pi^{\theta_l,\theta_e})$, the result of $\nabla_{\theta_l} f(\theta_l, \theta_e)$ is as follows:

$$\nabla_{\theta_l} f(\theta_l, \theta_e),$$

$$= E^{\pi^{\theta_l,\theta_e}} [\sum_{t=0}^{T-1} \gamma^t [(\mu_l(s^t, a^t) - \mu_l(s^t)) J_{ed}(s^t, a^t)^T]] + \mu_l(\pi^{\theta_l,\theta_e})$$

$$+ E^{\pi^{\theta_l,\theta_e}} [\sum_{t=0}^{T-1} \gamma^t (\mu_l(s^t, a^t) - \mu_l(s^t)) J_{ei}(s^t, a^t)^T],$$

$$\stackrel{(ii)}{=} E^{\pi^{\theta_l,\theta_e}} [\sum_{t=0}^{T-1} \gamma^t [(\mu_l(s^t, a^t) - \mu_l(s^t)) (J_{ed}(s^t, a^t) + J_{ei}(s^t, a^t))^T]] + \mu_l(\pi^{\theta_l,\theta_e}).$$

The $\nabla_{\theta_e} f(\theta_l, \theta_e)$ is calculated in the same way as the $\nabla_{\theta_l} f(\theta_l, \theta_e)$.

The process for getting $\nabla^2_{\theta_e} L(\theta_l, \theta_e)$ is shown below.
As we proved in Lemma 1, the gradient $\nabla_{\theta_e} L(\theta_l, \theta_e) = \mu_e(\pi^{\theta_l, \theta_e}) - \hat{\mu}_e(D_{r^{\theta_l}_{ld}, r_e}) + \lambda \theta_e$, as a result, we can take the derivative of each term separately.
The derivative of $\mu_e(\pi^{\theta_l, \theta_e})$ w.r.t $\theta_e$ is calculated as follows:

$$\nabla_{\theta_e} \mu_e(\pi^{\theta_l, \theta_e}),$$

$$\overset{(i)}{=} \int_{s^0 \in S} P_0(s^0) \int_{a^0_l \in a_l} \int_{a^0_e \in a_e} [\nabla_{\theta_e} \pi(a^0_l, a^0_e | s^0) \mu_e(s^0, a^0_l, a^0_e)^T$$

$$+ \pi(a^0_l, a^0_e | s^0) \nabla_{\theta_e} \mu_e(s^0, a^0_l, a^0_e)^T] da^0_l da^0_e ds^0,$$

$$= \int_{s^0 \in S} P_0(s^0) \int_{a^0_l \in a_l} \int_{a^0_e \in a_e} [\nabla_{\theta_l} \pi(a^0_l, a^0_e | s^0) \mu_e(s^0, a^0_l, a^0_e)^T$$

$$+ \pi(a^0_l, a^0_e | s^0)(\nabla^2_{\theta_e} r^{\theta_e}_e(s^0, a^0_l, a^0_e) + \gamma \int_{s^1 \in S} P(s^1 | s^0, a^0_l, a^0_e) \nabla_{\theta_e} \mu_e(s^1)) ds^1] da^0_l da^0_e ds^0,$$

$$\overset{(viii)}{=} \int_{s^0 \in S} P_0(s^0) \int_{a^0_l \in a_l} \int_{a^0_e \in a_e} \{\nabla_{\theta_l} \pi(a^0_l, a^0_e | s^0) \mu_e(s^0, a^0_l, a^0_e)^T$$

$$+ \pi(a^0_l, a^0_e | s^0)[\nabla^2_{\theta_e} r^{\theta_e}_e(s^0, a^0_l, a^0_e)$$

$$+ \gamma \int_{s^1 \in S} P(s^1 | s^0, a^0_l, a^0_e) \int_{a^1_l \in a_l} \int_{a^1_e \in a_e} \nabla_{\theta_e} \pi(a^1_l, a^1_e | s^1) \mu_e(s^1, a^1_l, a^1_e)^T$$

$$+ \pi(a^1_l, a^1_e | s^1)(\nabla^2_{\theta_e} r^{\theta_e}_e(s^1, a^1_l, a^1_e) + \gamma \int_{s^2 \in S} P(s^2 | s^1, a^1_l, a^1_e) \nabla_{\theta_e} \mu_e(s^2)) ds^2] ds^1 \} da^0_l da^0_e ds^0,$$

$$= E^{\pi^{\theta_l, \theta_e}} [\sum_{t=0}^{T-1} \gamma^t (\frac{\nabla_{\theta_e} \pi(a^t_l, a^t_e | s^t)}{\pi(a^t_l, a^t_e | s^t)} \mu_e(s^t, a^t)^T + \nabla^2_{\theta_e} r^{\theta_e}_e(s^t, a^t))],$$

$$= E^{\pi^{\theta_l, \theta_e}} [\sum_{t=0}^{T-1} \gamma^t (\nabla_{\theta_e} \ln(\pi^{\theta_l, \theta_e}) \mu_e(s^t, a^t)^T + \nabla^2_{\theta_e} r^{\theta_e}_e(s^t, a^t))],$$

$$\overset{(vii)}{=} E^{\pi^{\theta_l, \theta_e}} [\sum_{t=0}^{T-1} \gamma^t [(\mu_e(s^t, a^t) - \mu_e(s^t)) \mu_e(s^t, a^t)^T + \nabla^2_{\theta_e} r^{\theta_e}_e(s^t, a^t)]],$$

where $(vii)$ use the expression of the equation (10).
With the result of $\nabla_{\theta_e} \mu_e(\pi^{\theta_l, \theta_e})$, the derivative of $\nabla_{\theta_e} L(\theta_l, \theta_e)$ w.r.t $\theta_e$ is as follows:

$$\nabla^2_{\theta_e} L(\theta_l, \theta_e) = \nabla_{\theta_e}(\nabla_{\theta_e} L(\theta_l, \theta_e)),$$

$$= \nabla_{\theta_e}(\mu_e(\pi^{\theta_l, \theta_e}) - \hat{\mu}_e(D_{r^{\theta_l}_{ld}, r_e}) + \lambda \theta_e),$$

$$\overset{(ii)}{=} E^{\pi^{\theta_l, \theta_e}} [\sum_{t=0}^{T-1} \gamma^t [(\mu_e(s^t, a^t) - \mu_e(s^t)) \mu_e(s^t, a^t)^T + \nabla^2_{\theta_e} r^{\theta_e}_e(s^t, a^t)]]$$

$$- \frac{1}{d} \sum_{i=0}^{d} \sum_{t=0}^{T-1} \gamma^t \nabla^2_{\theta_e} r^{\theta_e}_e(s^{it}, a^{it}) + \lambda.$$

Through the same process of calculating $\nabla_{\theta_e} \mu_e(\pi^{\theta_l, \theta_e})$, the result is as follows:

$$\nabla^2_{\theta_l \theta_e} L(\theta_l, \theta_e) = E^{\pi^{\theta_l, \theta_e}} [\sum_{t=0}^{T-1} \gamma^t (\mu_e(s, a) - \mu_e(s)) \mu_l(s, a)^T]$$

$\square$

## A.5 Properties of the Lower Level Optimization Problem

**Lemma 4.** *Suppose Assumption 1 holds, for any $\theta_l \in \mathbb{R}^n$ and $\theta_e \in \mathbb{R}^m$, $L$ is continuously twice differentiable in $(\theta_l, \theta_e)$.*
*For any $\bar{\theta}_1 \in \mathbb{R}^n$, $\nabla_{\theta_e} L(\bar{\theta}_1, \theta_e)$ is Lipschitz continuous (w.r.t $\theta_e$) with constant $L_{L_{\theta_e}} > 0$.*
*For any $\bar{\theta}_1 \in \mathbb{R}^n$ and $\bar{\theta}_2 \in \mathbb{R}^m$, we have $\|\nabla^2_{\theta_l \theta_e} L(\bar{\theta}_1, \bar{\theta}_2)\| \leq C_{L_{\theta_l \theta_e}}$ for some constant $C_{L_{\theta_l \theta_e}} > 0$.*
*For any $\bar{\theta}_1 \in \mathbb{R}^n$, $\nabla^2_{\theta_l \theta_e} L(\bar{\theta}_1, \theta_e)$ and $\nabla^2_{\theta_e \theta_e} L(\bar{\theta}_1, \theta_e)$ are Lipschitz continuous (w.r.t $\theta_e$) with constants $L_{L_{\theta_l \theta_e}} > 0$ and $L_{L_{\theta_e \theta_e}} > 0$.*
*For any $\bar{\theta}_2 \in \mathbb{R}^m$, $\nabla^2_{\theta_l \theta_e} L(\theta_l, \bar{\theta}_2)$ and $\nabla^2_{\theta_e \theta_e} L(\theta_l, \bar{\theta}_2)$ are Lipschitz continuous (w.r.t $\theta_l$) with constants $\bar{L}_{L_{\theta_l \theta_e}} > 0$ and $\bar{L}_{L_{\theta_e \theta_e}} > 0$*

*Proof.* Suppose that $h$ is a real-valued function defined and differentiable on an interval $H \subset R^n$. If $\|\nabla h\|$ is bounded on I, then $h$ is a Lipschitz function on $H$. So we need to prove $\nabla^2_{\theta_e \theta_e} L(\theta_l, \theta_e)$ is bounded. From Assumption 1, we can show that $\exists R_{1g} > 0, \|\nabla r^{\theta_l}\| \leq R_{1g}$.

$$\|\mu_l(s)\| \leq E^{\pi^{\theta_l, \theta_e}}\left[\sum_{t=0}^{T-1} \gamma^t R_{1g} | s^0 = s\right] \overset{(i)}{\leq} \frac{R_{1g}}{1 - \gamma},$$

where $(i)$ uses the close form of geometric series.
As a result, $\|\mu_l(s)\|$ is bounded, proved through the same way, $\|\mu_e(s)\|, \|\mu_l(s, a)\|, \|\mu_e(s, a)\|$ are also bounded. Based on the Lemma 3, all elements of $\nabla^2_{\theta_e \theta_e} L(\theta_l, \theta_e)$ are finite, therefore, $\|\nabla^2_{\theta_e \theta_e} L(\theta_l, \theta_e)\|$ is bounded and the $\nabla_{\theta_e} L(\bar{\theta}_1, \theta_e)$ is Lipschitz continuous.
In the same way of proving $\|\nabla^2_{\theta_e \theta_e} L(\theta_l, \theta_e)\|$ is bounded, we can show $\|\nabla^2_{\theta_l \theta_e} L(\theta_l, \theta_e)\|$ is bounded.

We need to prove $\nabla^3_{\theta_e \theta_e \theta_e} L(\theta_l, \theta_e)$ and $\nabla^3_{\theta_l \theta_e \theta_e} L(\theta_l, \theta_e)$ are bounded. The proof of $\nabla^3_{\theta_l \theta_e \theta_e} L(\theta_l, \theta_e)$ is bounded as follows:

$$\nabla^3_{\theta_l \theta_e \theta_e} L(\theta_l, \theta_e),$$

$$= \nabla_{\theta_e}(E^{\pi^{\theta_l, \theta_e}}\left[\sum_{t=0}^{T-1} \gamma^t (\mu_e(s, a) - \mu_e(s))\mu_l(s, a)^T\right]),$$

$$\overset{(i)}{=} E^{\pi^{\theta_l, \theta_e}}\left[\sum_{t=0}^{T-1} \gamma^t (\nabla_{\theta_e}\mu_e(s, a))\mu_l(s, a)^T + \mu_e(s, a)(\nabla_{\theta_e}\mu_l(s, a)^T)\right.$$

$$\left. - (\nabla_{\theta_e}\mu_e(s))\mu_l(s, a)^T - \mu_e(s)(\nabla_{\theta_e}\mu_l(s, a)^T)\right].$$

Each gradient inside the expectation could be derived through the process of deriving $\nabla_{\theta_l}\mu_l(\pi^{\theta_l, \theta_e})$ in the proof of Lemma 3 and these gradients are all finite with the same way of proving $\|\nabla^2_{\theta_e \theta_e} L(\theta_l, \theta_e)\|$ is bounded. The third-order gradient $\nabla^3_{\theta_l \theta_e \theta_e} L(\theta_l, \theta_e)$ is bounded with the smae process. Therefore, the third-order gradients of $L(\theta_l, \theta_e)$ are all bounded.
Through the same procedure, we can prove $\nabla^3_{\theta_e \theta_e \theta_l} L(\theta_l, \theta_e)$ and $\nabla^3_{\theta_l \theta_e \theta_l} L(\theta_l, \theta_e)$ are bounded.
At the same time, the existence of $\nabla^2_{\theta_l \theta_e} L(\theta_l, \theta_e), \nabla^2_{\theta_e \theta_e} L(\theta_l, \theta_e)$ are shown. Analogously, the existence of $\nabla^2_{\theta_l \theta_l} L(\theta_l, \theta_e), \nabla^2_{\theta_e \theta_l} L(\theta_l, \theta_e)$ could be proved in the same way. The third-order gradients of $L(\theta_l, \theta_e)$ are bounded. Therefore, $L$ is continuously twice differentiable in $(\theta_l, \theta_e)$

$\square$

## A.6 Properties of the Upper-Level Optimization Problem

**Lemma 5.** *Suppose Assumption 1 holds, for any $\bar{\theta}_1 \in \mathbb{R}^n$, $\nabla_{\theta_l} f(\bar{\theta}_1; \theta_e)$ and $\nabla_{\theta_e} f(\bar{\theta}_1; \theta_e)$ are Lipschitz continuous (w.r.t $\theta_e$) with constants $L_{f_{\theta_l}} > 0$ and $L_{f_{\theta_e}} > 0$.*
*For any $\bar{\theta}_2 \in \mathbb{R}^m$, $\nabla_{\theta_e} f(\theta_l; \bar{\theta}_2)$ is Lipschitz continuous (w.r.t $\theta_l$) with constants $\bar{L}_{f_{\theta_e}} > 0$*

*For any $\bar{\theta}_1 \in \mathbb{R}^n$ and $\bar{\theta}_2 \in \mathbb{R}^m$, we have $\|\nabla_{\theta_e} f(\bar{\theta}_1; \bar{\theta}_2)\| \leq C_{f_{\theta_e}}$ for some $C_{f_{\theta_e}} > 0$.*

*Proof.*

$$\|J_{ed}(\pi^{\theta_l, \theta_e})\| \leq E^{\pi^{\theta_l, \theta_e}}[\sum_{t=0}^{T-1} \gamma^t r_{ld}|s^0 = s] \overset{(i)}{\leq} \frac{r_{ld}}{1-\gamma},$$

where $(i)$ uses the close form of geometric series.

So the cumulative expert-dependent reward value $J_{ed}$ is bounded, analogously, the cumulative expert-independent reward value $J_{ei}$ is bounded.

$$\nabla_{\theta_l \theta_e} f(\theta_l; \theta_e),$$

$$= \nabla_{\theta_e}(E^{\pi^{\theta_l, \theta_e}}[\sum_{t=0}^{T-1} \gamma^t[(\mu_l(s^t, a^t) - \mu_l(s^t))(J_{ed}(s^t, a^t) + J_{ei}(s^t, a^t))^T]] + \mu_l(\pi^{\theta_l, \theta_e}),$$

$$\overset{(i)}{=} E^{\pi^{\theta_l, \theta_e}}[\sum_{t=0}^{T-1} \gamma^t[(\nabla_{\theta_e}\mu_l(s^t, a^t) - \nabla_{\theta_e}\mu_l(s^t))(J_{ed}(s^t, a^t) + J_{ei}(s^t, a^t))^T$$

$$+ (\mu_l(s^t, a^t) - \mu_l(s^t))(\nabla_{\theta_e}J_{ed}(s^t, a^t) + \nabla_{\theta_e}J_{ei}(s^t, a^t))^T]] + \nabla_{\theta_e}\mu_l(\pi^{\theta_l, \theta_e}).$$

Refer to the proof of Lemma 4, all elements in $\nabla_{\theta_l \theta_e} f(\theta_l; \theta_e)$ are finite, itself is bounded. Analogously, $\nabla_{\theta_e \theta_e} f(\theta_l; \theta_e)$ is bounded under the same proofing process.

Through the same procedure , we can prove $\nabla_{\theta_e \theta_l} f(\theta_l; \theta_e)$ is bounded.

All element of $\nabla_{\theta_e} f(\bar{\theta}_1; \bar{\theta}_2)$ are bounded. Therefore, all elements for $\nabla_{\theta_e} f(\theta_l; \theta_e)$ is bounded and it is bounded.

$\square$

## A.7 PROPERTIES OF THE APPROXIMATION ERRORS

Define $b_{12}(k) = \hat{\nabla}^2_{\theta_l \theta_e} L(\theta_l(k), \theta_e(k)) - \nabla^2_{\theta_l \theta_e} L(\theta_l(k), \theta_e(k))$, $b_{22}(k) = \hat{\nabla}^2_{\theta_e \theta_e} L(\theta_l(k), \theta_e(k)) - \nabla^2_{\theta_e \theta_e} L(\theta_l(k), \theta_e(k))$, $b_1(k) = \hat{\nabla}_{\theta_l} f(\theta_l(k), \theta_e(k)) - \nabla_{\theta_l} f(\theta_l(k), \theta_e(k))$, $b_2(k) = \hat{\nabla}_{\theta_e} f(\theta_l(k), \theta_e(k)) - \nabla_{\theta_e} f(\theta_l(k), \theta_e(k))$, $b(k) = [\hat{\nabla}^2_{\theta_e \theta_e} L(\theta_l(k), \theta_e(k))]^{-1}\hat{\nabla}_{\theta_e} f(\theta_l(k), \theta_e(k)) - [\nabla^2_{\theta_e \theta_e} L(\theta_l(k), \theta_e(k))]^{-1}\nabla_{\theta_e} f(\theta_l(k), \theta_e(k))$ at iteration $k$, $b_a(k) = \hat{\nabla} f(\theta_l(k), \theta_e(k)) - \nabla f(\theta_l(k), \theta_e(k))$ at the iteration $k$. Through the same procedure in the proof of the Lemma 5, we can get conclude that each element of forth-order gradient of $L(\theta_l, \theta_e)$ is bounded by the constant $C_L$ and each element of the third-order gradient of $f(\theta_l(k), \theta_e(k))$ is bounded by $C_f$.

**Lemma 6.** *The biases $b_{12}(k)$, $b_{22}(k)$, $b_1(k)$, $b_2(k)$, $b(k)$, and $b_a(k)$ are bounded,*

$$\|b_{12}(k)\| \leq \frac{C_L p^2(k)\sqrt{m}}{6}\{[m^3 - (m-1)^3]\alpha_l^2 + (m-1)^3\alpha_l\alpha_0^3\},$$

$$\|b_{22}(k)\| \leq \frac{C_L p^2(k)\sqrt{m}}{6}\{[m^3 - (m-1)^3]\alpha_l^2 + (m-1)^3\alpha_l\alpha_0^3\},$$

$$\|b_1(k)\| \leq \frac{C_f p^2(k)\sqrt{n}}{6}\{[n^3 - (n-1)^3]\alpha_l^2 + (n-1)^3\alpha_l\alpha_0^3\},$$

$$\|b_2(k)\| \leq \frac{C_f p^2(k)\sqrt{m}}{6}\{[m^3 - (m-1)^3]\alpha_l^2 + (m-1)^3\alpha_l\alpha_0^3\},$$

$$\|b(k)\| \leq \frac{2C_{f_{\theta_e}}(C_L p^2(k) + C_f p^2(k))}{6\sqrt{m}\lambda^2}\{[m^3 - (m-1)^3]\alpha_l^2 + (m-1)^3\alpha_l\alpha_0^3\},$$

$$\|b_a(k)\|$$

$$\leq \frac{C_f p^2(k)\sqrt{n}}{6}\{[n^3 - (n-1)^3]\alpha_l^2 + (n-1)^3\alpha_l\alpha_0^3\}$$

$$+ \frac{2C_{L_{\theta_l\theta_e}}C_L p^2(k) + 2C_{L_{\theta_l\theta_e}}\sqrt{m}\lambda C_f p^2(k)}{6\sqrt{m}\lambda^2}$$

$$\{[m^3 - (m-1)^3]\alpha_l^2 + (m-1)^3\alpha_l\alpha_0^3\} + \frac{C_{f_{\theta_e}}C_L p^2(k)}{6m\lambda^3}\{[m^3 - (m-1)^3]\alpha_l^2 + (m-1)^3\alpha_l\alpha_0^3\}$$

$$+ \frac{2p^4(k)C_L(C_{f_{\theta_e}}C_L + \sqrt{m}\lambda C_f)}{36\lambda^2}\{[m^3 - (m-1)^3]\alpha_l^2 + (m-1)^3\alpha_l\alpha_0^3\}^2,$$

*Proof.* According to the Lemma 1 in (Spall, 1992), the approximation error $b_{12l}(k)$ for $\hat{\nabla}^2_{\theta_l\theta_e}L(\theta_l, \theta_e)$ is :

$$b_{12l}(k) \leq \frac{C_L p^2(k)}{6}\{[m^3 - (m-1)^3]\alpha_l^2 + (m-1)^3\alpha_l\alpha_0^3\},$$

where $b_{12l}(k)$ represent the $l$th term of the bias $b_{12l}(k)$ at $k$th iteration.

$$\|b_{12}(k)\| \leq \frac{C_L p^2(k)\sqrt{m}}{6}\{[m^3 - (m-1)^3]\alpha_l^2 + (m-1)^3\alpha_l\alpha_0^3\}.$$

Analogously, the $b_{22}(k)$, $b_1(k)$ and $b_2(k)$ are also bounded.

With the $\hat{\nabla}^2_{\theta_e\theta_e}L(\theta_l(k), \theta_e(k))$ and $\hat{\nabla}_{\theta_e}f(\theta_l(k), \theta_e(k))$, we estimate the $[\nabla^2_{\theta_e\theta_e}L(\theta_l(k), \theta_e(k))]^{-1}\nabla_{\theta_e}f(\theta_l(k), \theta_e(k))$ through conjugate gradient.

$$\|\hat{\nabla}^2_{\theta_e\theta_e}L(\theta_l(k), \theta_e(k))\| = \|\nabla^2_{\theta_e\theta_e}L(\theta_l(k), \theta_e(k)) + b_{22}(k)\| \geq \|\lambda I + b_{22}(k)\|.$$

Then we can tune the parameters of $b_{22}(k)$ such as $\Delta$, $\alpha_0$ and $\alpha_l$ to let $\|[\hat{\nabla}^2_{\theta_e\theta_e}L(\theta_l, \theta_e)]^{-1}\| \leq \frac{2}{\sqrt{m}\lambda}$ and make sure the $\hat{\nabla}^2_{\theta_e\theta_e}L(\theta_l, \theta_e) \geq \frac{\lambda}{2}I$

$$\|b(k)\|,$$
$$= \|E[[\hat{\nabla}^2_{\theta_e\theta_e}L(\theta_l(k), \theta_e(k))]^{-1}\hat{\nabla}_{\theta_e}f(\theta_l(k), \theta_e(k))$$
$$- [\nabla^2_{\theta_e\theta_e}L(\theta_l(k), \theta_e(k))]^{-1}\nabla_{\theta_e}f(\theta_l(k), \theta_e(k))]\|,$$
$$\overset{(iv)}{\leq} \|E[\hat{\nabla}^2_{\theta_e\theta_e}L(\theta_l(k), \theta_e(k))]^{-1}(\hat{\nabla}^2_{\theta_e\theta_e}L(\theta_l(k), \theta_e(k)) - \nabla^2_{\theta_e\theta_e}L(\theta_l(k), \theta_e(k)))$$
$$[\nabla^2_{\theta_e\theta_e}L(\theta_l(k), \theta_e(k))]^{-1}]\|\|\nabla_{\theta_e}f(\theta_l(k), \theta_e(k))\| + \|\hat{\nabla}^2_{\theta_e\theta_e}L(\theta_l(k), \theta_e(k))]^{-1}b_2(k)\|,$$
$$\overset{(v)}{\leq} \|[\hat{\nabla}^2_{\theta_e\theta_e}L(\theta_l(k), \theta_e(k))]^{-1}\|\|\hat{\nabla}^2_{\theta_e\theta_e}L(\theta_l(k), \theta_e(k)) - \nabla^2_{\theta_e\theta_e}L(\theta_l(k), \theta_e(k))\|\|$$
$$[\nabla^2_{\theta_e\theta_e}L(\theta_l(k), \theta_e(k))]^{-1}\|\|\nabla_{\theta_e}f(\theta_l(k), \theta_e(k))\| + \|\hat{\nabla}^2_{\theta_e\theta_e}L(\theta_l(k), \theta_e(k))]^{-1}\|\|b_2 k\|,$$
$$\overset{(vii)}{\leq} \frac{2C_{f_{\theta_e}}\|b_{22}(k)\| + 2\sqrt{m}\lambda\|b_2(k)\|}{m\lambda^2},$$
$$\leq \frac{2C_{f_{\theta_e}}(C_L p^2(k) + C_f p^2(k))}{6\sqrt{m}\lambda^2}\{[m^3 - (m-1)^3]\alpha_l^2 + (m-1)^3\alpha_l\alpha_0^3\},$$

where $(vii)$ uses the result of Lemma (5).

$$\|b_a(k)\| = E[\|\hat{\nabla} f(\theta_l(k), \theta_e(k))\|] - \|\nabla f(\theta_l(k), \theta_e(k))\|,$$

$$\overset{(vii)}{=} E[\|\hat{\nabla}_{\theta_l} f(\theta_l(k), \theta_e(k)) - \hat{\nabla}^2_{\theta_l \theta_e} L(\theta_l(k), \theta_e(k)) [\hat{\nabla}^2_{\theta_e \theta_e} L(\theta_l(k), \theta_e(k))]^{-1}$$

$$\hat{\nabla}_{\theta_e} f(\theta_l(k), \theta_e(k))\|] - \|\nabla f(\theta_l(k), \theta_e(k))\|,$$

$$= \|\nabla_{\theta_l} f(\theta_l(k), \theta_e(k)) + b_1(k) - (\nabla^2_{\theta_l \theta_e} L(\theta_l(k), \theta_e(k)) + b_{12}(k))$$

$$([\nabla^2_{\theta_e \theta_e} L(\theta_l(k), \theta_e(k))]^{-1} \nabla_{\theta_e} f(\theta_l(k), \theta_e(k)) + b(k))\| - \|\nabla f(\theta_l(k), \theta_e(k))\|$$

$$\overset{(v)}{\leq} \|b_1(k)\| + \|\nabla^2_{\theta_l \theta_e} L(\theta_l(k), \theta_e(k))\| \|b(k)\|$$

$$+ \|[\nabla^2_{\theta_e \theta_e} L(\theta_l(k), \theta_e(k))]^{-1}\| \|\nabla_{\theta_e} f(\theta_l(k), \theta_e(k))\| \|b_{12}(k)\| + \|b_{12}(k)\| \|b(k)\|,$$

$$\overset{(vii)}{\leq} \frac{C_f p^2(k) \sqrt{n}}{6} \{[n^3 - (n-1)^3]\alpha_l^2 + (n-1)^3 \alpha_l \alpha_0^3\}$$

$$+ \frac{2C_{L_{\theta_l \theta_e}} C_L p^2(k) + 2C_{L_{\theta_l \theta_e}} \sqrt{m} \lambda C_f p^2(k)}{6\sqrt{m}\lambda^2} \{[m^3 - (m-1)^3]\alpha_l^2 + (m-1)^3 \alpha_l \alpha_0^3\}$$

$$+ \frac{C_{f_{\theta_e}} C_L p^2(k)}{6m\lambda^3} \{[m^3 - (m-1)^3]\alpha_l^2 + (m-1)^3 \alpha_l \alpha_0^3\}$$

$$+ \frac{2p^4(k) C_L (C_{f_{\theta_e}} C_L + \sqrt{m}\lambda C_f)}{36\lambda^2} \{[m^3 - (m-1)^3]\alpha_l^2 + (m-1)^3 \alpha_l \alpha_0^3\}^2,$$

where the first $(vii)$ uses the result of Lemma (2), and the second $(vii)$ uses the result of Lemma (5). $\qquad \square$

## A.8 PROOF OF THEOREM 2

From the Lemma 2.2 of (Ghadimi & Wang, 2018), $\|\nabla f(\theta_l(k), \theta_e(k)) - \nabla f(\theta_l(k), \theta_e^*(\theta_l(k)))\| \leq C\|\theta_e^*(\theta_l(k)) - \theta_e(k)\|$, $\|\nabla f(\theta_l(k'), \theta_e^*(\theta_l(k'))) - \nabla f(\theta_l(k), \theta_e^*(\theta_l(k)))\| \leq L_f \|\theta_l(k') - \theta_l(k)\|$ where $C = L_{f_{\theta_l}} + \frac{L_{f_{\theta_e}} C_{L_{\theta_l \theta_e}}}{\lambda} + C_{f_{\theta_e}}[\frac{L_{L_{\theta_l \theta_e}}}{\lambda} + \frac{L_{L_{\theta_e \theta_e}} C_{L_{\theta_l \theta_e}}}{\lambda^2}]$, $L_f = \frac{(\bar{L}_{f_{\theta_e}} + C)C_{L_{\theta_l \theta_e}}}{\lambda} + L_{f_{\theta_l}} + C_{f_{\theta_e}}[\frac{\bar{L}_{L_{\theta_l \theta_e}} C_{f_{\theta_e}}}{\lambda} + \frac{\bar{L}_{L_{\theta_e \theta_e}} C_{L_{\theta_l \theta_e}}}{\lambda^2}]$, $Q_L = \frac{L_{L_{\theta_e}}}{\lambda}$ denotes the condition number of $L(\theta_l, \theta_e)$, $M = \max_{\theta_l \in \theta_l} \|\theta_e(0) - \theta_e^*(\theta_l)\|$, $D_{\theta_l} = \max_{x,y \in \theta_l}\{\|x - y\|\}$.
The proof of Theorem 2 is as follows:

*Proof.* First we compute the variance of $\hat{\nabla} f(\theta_l(k), \theta_e(k))$.

$\|\hat{\nabla} f(\theta_l(k), \theta_e(k))\|,$

$\overset{(vii)}{=} \|\hat{\nabla}_{\theta_l} f(\theta_l(k), \theta_e(k)) - \hat{\nabla}^2_{\theta_l \theta_e} L(\theta_l(k), \theta_e(k)) [\hat{\nabla}^2_{\theta_e \theta_e} L(\theta_l(k), \theta_e(k))]^{-1} \hat{\nabla}_{\theta_e} f(\theta_l(k), \theta_e(k))\|,$

$= \|\nabla_{\theta_l} f(\theta_l(k), \theta_e(k)) + b_1(k) - (\nabla^2_{\theta_l \theta_e} L(\theta_l(k), \theta_e(k)) + b_{12}(k))$

$([\nabla^2_{\theta_e \theta_e} L(\theta_l(k), \theta_e(k))]^{-1} \nabla_{\theta_e} f(\theta_l(k), \theta_e(k)) + b(k))\|,$

$\overset{(v)}{\leq} \|\nabla_{\theta_l} f(\theta_l(k), \theta_e(k))\| + \|b_1(k)\| + \|\nabla^2_{\theta_l \theta_e} L(\theta_l(k), \theta_e(k))\| \|[\nabla^2_{\theta_e \theta_e} L(\theta_l(k), \theta_e(k))]^{-1}\|$

$\|\nabla_{\theta_e} f(\theta_l(k), \theta_e(k))\| + \|\nabla^2_{\theta_l \theta_e} L(\theta_l(k), \theta_e(k))\| \|b(k)\| + \|b_1(k)\| \|[\nabla^2_{\theta_e \theta_e} L(\theta_l(k), \theta_e(k))]^{-1}\|$

$\|\nabla_{\theta_e} f(\theta_l(k), \theta_e(k))\| + \|b_1(k)\| \|b(k)\|,$

$\overset{(vii)}{\leq} C_{f_{\theta_l}} + \frac{C_f p^2(k) \sqrt{n}}{6} \{[n^3 - (n-1)^3] \alpha_l^2 + (n-1)^3 \alpha_l \alpha_0^3\}$

$+ \frac{C_{L_{\theta_l \theta_e}} C_{f_{\theta_e}}}{\sqrt{m} \lambda} + \frac{2 C_{L_{\theta_l \theta_e}} C_{f_{\theta_e}} (C_L p^2(k) + C_f p^2(k))}{6 \sqrt{m} \lambda^2} \{[m^3 - (m-1)^3] \alpha_l^2 + (m-1)^3 \alpha_l \alpha_0^3\}$

$+ \frac{C_{f_{\theta_e}}}{\sqrt{m} \lambda} \frac{C_f p^2(k) \sqrt{n}}{6} \{[n^3 - (n-1)^3] \alpha_l^2 + (n-1)^3 \alpha_l \alpha_0^3\}$

$+ \frac{2 C_f p^2(k) \sqrt{n} C_{f_{\theta_e}} (C_L p^2(k) + C_f p^2(k))}{36 \sqrt{m} \lambda^2}$

$\{[n^3 - (n-1)^3] \alpha_l^2 + (n-1)^3 \alpha_l \alpha_0^3\} \{[m^3 - (m-1)^3] \alpha_l^2 + (m-1)^3 \alpha_l \alpha_0^3\},$

where the first $(vii)$ uses the result of Lemma (2), and the second $(vii)$ uses the result of Lemma (5) and Lemma (6).

Since $\hat{\nabla} f(\theta_l(k), \theta_e(k))$ is bounded,

$Var(\hat{\nabla} f(\theta_l(k), \theta_e(k))),$

$\leq (C_{f_{\theta_l}} + \frac{C_f p^2(k) \sqrt{n}}{6} \{[n^3 - (n-1)^3] \alpha_l^2 + (n-1)^3 \alpha_l \alpha_0^3\}$

$+ \frac{C_{L_{\theta_l \theta_e}} C_{f_{\theta_e}}}{\sqrt{m} \lambda} + \frac{2 C_{L_{\theta_l \theta_e}} C_{f_{\theta_e}} (C_L p^2(k) + C_f p^2(k))}{6 \sqrt{m} \lambda^2} \{[m^3 - (m-1)^3] \alpha_l^2 + (m-1)^3 \alpha_l \alpha_0^3\}$

$+ \frac{C_{f_{\theta_e}}}{\sqrt{m} \lambda} \frac{C_f p^2(k) \sqrt{n}}{6} \{[n^3 - (n-1)^3] \alpha_l^2 + (n-1)^3 \alpha_l \alpha_0^3\}$

$+ \frac{2 C_f p^2(k) \sqrt{n} C_{f_{\theta_e}} (C_L p^2(k) + C_f p^2(k))}{36 \sqrt{m} \lambda^2}$

$\{[n^3 - (n-1)^3] \alpha_l^2 + (n-1)^3 \alpha_l \alpha_0^3\} \{[m^3 - (m-1)^3] \alpha_l^2 + (m-1)^3 \alpha_l \alpha_0^3\})^2.$

Then we need to find the total bias, the total bias $b_t(k)$ is the sum of the bias from approximation and $\nabla f(\theta_l(k), \theta_e(k)) - \nabla f(\theta_l(k), \theta_e^*(\theta_l(k)))$

$\|b_t(k)\|,$

$= \|E[\nabla f(\theta_l(k), \theta_e(k))] - \nabla f(\theta_l(k), \theta_e^*(k))\|,$

$\overset{(vii)}{\leq} \|b_a(k)\| + \|r_{li} (\frac{Q_L - 1}{Q_L + 1})^{t_k} \|\theta_e(0) - \theta_e^*(\theta_l(k))\|,$

where $(vii)$ adds the approximation error $r_{li} (\frac{Q_L - 1}{Q_L + 1})^{t_k} \|\theta_e(0) - \theta_e^*(\theta_l(k))$ from the lower level. Next step is to find the bound for $E[\|\nabla f(\theta_l(k), \theta_e^*(\theta_l(k)))\|^2]$.

$f(\theta_l(k+1), \theta_e^*(\theta_l(k+1))),$

$\overset{(vi)}{\leq} f(\theta_l(k), \theta_e^*(\theta_l(k))) + \langle \nabla f(\theta_l(k), \theta_e^*(\theta_l(k))), \theta_l(k+1) - \theta_l(k) \rangle + \frac{L_f}{2} \|\theta_l(k+1) - \theta_l(k)\|^2,$

$= f(\theta_l(k), \theta_e^*(\theta_l(k))) - \alpha_k \langle \nabla f(\theta_l(k), \theta_e^*(\theta_l(k))), \hat{\nabla} f(\theta_l(k), \theta_e(k)) \rangle + \frac{L_f \alpha_k^2}{2} \|\hat{\nabla} f(\theta_l(k), \theta_e(k))\|^2.$

The expectation of $f(\theta_l(k+1), \theta_e^*(\theta_l(k+1)))$ becomes:

$$E[f(\theta_l(k+1), \theta_e^*(\theta_l(k+1)))],$$

$$\leq f(\theta_l(k), \theta_e^*(\theta_l(k))) - \alpha_k \langle \nabla f(\theta_l(k), \theta_e^*(\theta_l(k))), \nabla f(\theta_l(k), \theta_e^*(\theta_l(k))) + b_t(k) \rangle$$

$$+ \frac{L_f \alpha_k^2}{2} E \| \nabla f(\theta_l(k), \theta_e^*(\theta_l(k))) + \hat{\nabla} f(\theta_l(k), \theta_e(k)) - \nabla f(\theta_l(k), \theta_e^*(\theta_l(k))) \|^2,$$

$$\leq f(\theta_l(k), \theta_e^*(\theta_l(k))) - \alpha_k \langle \nabla f(\theta_l(k), \theta_e^*(\theta_l(k))), \nabla f(\theta_l(k), \theta_e^*(\theta_l(k))) + b_t(k) \rangle$$

$$+ \frac{L_f \alpha_k^2}{2} Var(\hat{\nabla} f(\theta_l(k), \theta_e(k))) + \frac{L_f \alpha_k^2}{2} E \| \nabla f(\theta_l(k), \theta_e^*(\theta_l(k))) \|^2$$

$$+ L_f \alpha_k^2 \langle \nabla f(\theta_l(k), \theta_e^*(\theta_l(k))), b_t(k) \rangle,$$

$$= f(\theta_l(k), \theta_e^*(\theta_l(k))) - (\alpha_k - \frac{L_f \alpha_k^2}{2}) \| \nabla f(\theta_l(k), \theta_e^*(\theta_l(k))) \|^2$$

$$- (\alpha_k - L_f \alpha_k^2) \langle \nabla f(\theta_l(k), \theta_e^*(\theta_l(k))), b_t(k) \rangle + \frac{L_f \alpha_k^2}{2} Var(\hat{\nabla} f(\theta_l(k), \theta_e(k))) + \frac{L_f \alpha_k^2}{2} \| b_t(k) \|^2.$$

Choose $\alpha_k \leq \frac{1}{L_f}$ and with the fact $2 \langle \nabla f(\theta_l(k), \theta_e^*(\theta_l(k))), b_t(k) \rangle \leq \| \nabla f(\theta_l(k), \theta_e^*(\theta_l(k))) \|^2 + \| b_t(k) \|^2$.

$$E[f(\theta_l(k+1), \theta_e^*(\theta_l(k+1)))],$$

$$\leq f(\theta_l(k), \theta_e^*(\theta_l(k))) - \frac{\alpha_k}{2} \| \nabla f(\theta_l(k), \theta_e^*(\theta_l(k))) \|^2$$

$$+ \frac{\alpha_k}{2} \| b_t(k) \|^2 + \frac{L_f \alpha_k^2}{2} Var(\hat{\nabla} f(\theta_l(k), \theta_e(k))),$$

Rearrange terms,

$$\sum_{k=0}^{K-1} \frac{\alpha_k}{2} E[\| \nabla f(\theta_l(k), \theta_e^*(\theta_l(k))) \|^2],$$

$$\leq f(\theta_l(0), \theta_e^*(\theta_l(0))) - f^* + \sum_{k=0}^{K-1} (\frac{\alpha_k}{2} \| b_t(k) \|^2 + \frac{L_f \alpha_k^2}{2} Var(\hat{\nabla} f(\theta_l(k), \theta_e(k)))).$$

For $\| b_t(k) \|^2$, it is the linear combination of $p^4(k), p^6(k), p^8(k), p^2(k)(\frac{Q_L-1}{Q_L+1})^{t_k} +, p^4(k)(\frac{Q_L-1}{Q_L+1})^{t_k}, (\frac{Q_L-1}{Q_L+1})^{2t_k}$. For $Var(\hat{\nabla} f(\theta_l(k), \theta_e(k)))$, it is the linear combination of $1, p^2(k), p^4(k), p^6(k), p^8(k)$. For simplification, we use $C_{si} > 0, i = 1, 2, \ldots$ to represent the constant for all combinations of terms involve $p(k), \alpha_k$ and $t_k$. Then we can continue the calculation as follows:

$$\frac{1}{K} \sum_{k=0}^{K-1} E[\| \nabla f(\theta_l(k), \theta_e^*(\theta_l(k))) \|^2],$$

$$\leq \frac{2}{K\alpha_k} f(\theta_l(0), \theta_e^*(\theta_l(0))) - f^* + \frac{1}{K} \sum_{k=0}^{K-1} (\| b_t(k) \|^2 + L_f \alpha_k Var(\hat{\nabla} f(\theta_l(k), \theta_e(k)))),$$

$$\overset{(viii)}{\leq} \frac{2(f(\theta_l(0), \theta_e^*(\theta_l(0))) - f^*)}{\alpha_k K} + \frac{C_{s1}}{K} \sum_{k=0}^{K-1} p^4(k) + \frac{C_{s2}}{K} \sum_{k=0}^{K-1} p^6(k) + \frac{C_{s3}}{K} \sum_{k=0}^{K-1} p^8(k)$$

$$+ \frac{C_{s4}}{K} \sum_{k=0}^{K-1} p^2(k)(\frac{Q_L-1}{Q_L+1})^{t_k} + \frac{C_{s5}}{K} \sum_{k=0}^{K-1} p^4(k)(\frac{Q_L-1}{Q_L+1})^{t_k} + \frac{C_{s6}}{K} \sum_{k=0}^{K-1} (\frac{Q_L-1}{Q_L+1})^{2t_k}$$

$$+ \frac{C_{s7}}{K} \sum_{k=0}^{K-1} \alpha_k + \frac{C_{s8}}{K} \sum_{k=0}^{K-1} \alpha_k p^2(k) + \frac{C_{s9}}{K} \sum_{k=0}^{K-1} \alpha_k p^4(k) + \frac{C_{s10}}{K} \sum_{k=0}^{K-1} \alpha_k p^6(k)$$

$$+ \frac{C_{s11}}{K} \sum_{k=0}^{K-1} \alpha_k p^8(k),$$

where $(viii)$ expands each term of $\| b_t(k) \|^2$ and $Var(\hat{\nabla} f(\theta_l(k), \theta_e(k)))$. Choose $p(k) = \frac{1}{k}$, $\alpha_k = \frac{1}{L_f \sqrt{K}}$, $t_k = \lceil \frac{\sqrt[4]{k+1}}{2} \rceil$. Since $0 \leq \frac{Q_L-1}{Q_L+1} < 1$, we can conclude $\sum_{k=0}^{K-1} p^2(k)(\frac{Q_L-1}{Q_L+1})^{t_k} <$

$\sum_{k=0}^{K-1} p^2(k)$ when $\frac{Q_L-1}{Q_L+1} \neq 0$, then the convergence rate is as follows:

$$\frac{1}{K} \sum_{k=0}^{K-1} E[\|\nabla f(\theta_l(k), \theta_e^*(\theta_l(k)))\|^2] \leq \frac{C_{s12}}{\sqrt{K}} + \frac{C_{s13}}{K} + \frac{C_{s14}}{K\sqrt{K}}.$$

As $K \to \infty$, $\frac{1}{K} \sum_{k=0}^{K-1} E[\|\nabla f(\theta_l(k), \theta_e^*(\theta_l(k)))\|^2] \to 0$, which shows that $E[\nabla f(\theta_l(k), \theta_e^*(\theta_l(k)))]$ decreases at the rate of $\mathcal{O}(\frac{1}{\sqrt{K}} + \frac{1}{K} + \frac{1}{K\sqrt{K}})$. □

### A.8.1 PROOF OF COROLLARY 2.1

Define the cumulative reward function of the expert as $J_e(\pi) \triangleq E^\pi[\sum_{t=0}^{T-1} \gamma^t r_e^{\theta_e}(s^t, a^t)]$. If $r_e^{\theta_e}$ is a linear reward function, we have $r_e^{\theta_e} \triangleq \langle \theta_e, \phi(s, a_e) \rangle$ where the feature $\phi(s, a_e) \in \mathbb{R}^{d_{\theta_e}}$ and $d_{\theta_e}$ is the dimension of $\theta_e$. Then the expert's feature expectation is formulated as $\mu_f(\pi) \triangleq E^\pi[\sum_{t=0}^{T-1} \gamma^t \phi(s^t, a_e^t)]$. From Theorem 2, we can get $\|\mu_f(\pi_{\theta_e}) - \mu_f(\pi_e)\|^2$ decreases in $\mathcal{O}(\frac{1}{\sqrt{K}})$.

*Proof.*

$$
\begin{aligned}
J_e(\pi_{\theta_e}) - J_e(\pi_e) &= \langle \theta_e, \mu_f(\pi_{\theta_e}) - \mu_f(\pi_e) \rangle \\
&\leq \max_{\theta_e \in \Theta_e} \langle \theta_e, \mu_f(\pi_{\theta_e}) - \mu_f(\pi_e) \rangle \\
&\stackrel{(v)}{\leq} \max_{\theta_e \in \Theta_e} \|\theta_e\| \|\mu_f(\pi_{\theta_e}) - \mu_f(\pi_e)\|, \\
&= \|\mu_f(\pi_{\theta_e}) - \mu_f(\pi_e)\|, \\
&\stackrel{(vii)}{\leq} \frac{C_{15}}{\sqrt[4]{K}},
\end{aligned}
$$

where $(vii)$ uses the result $\|\mu_f(\pi_{\theta_e}) - \mu_f(\pi_e)\|^2$ decreases in $\mathcal{O}(\frac{1}{\sqrt{K}})$ and $C_{15}$ is the constant number which includes all influence factors other than $K$. □

### A.9 PROOF OF THEOREM 1

The prove of Theorem 1 is as follows: According to the Algorithm 1 in (Ziebart et al., 2008), we need to compute the state frequency for the $\mu_e(\pi^{\theta_l, \theta_e})$. For each state-action pair, it needs to recursively compute for up to $T$ iterations. As there are $T$ state-action pairs in one demonstration, the computation complexity for the $\mu_e(\pi^{\theta_l, \theta_e})$ is $\mathcal{O}(T^2)$ is we see $T$ as the deterministic factor. Analogously, the computational complexity for $\mu_l(\pi^{\theta_l, \theta_e}), \mu_e(s, a), \mu_e(s), \mu_l(s, a), \mu_l(s), J_{ei}(s, a), J_{ed}(\pi^{\theta_l, \theta_e}), J_{ei}(\pi^{\theta_l, \theta_e})$ are $\mathcal{O}(T^2)$.

For SPSA, the required terms are $f(\theta_l, \theta_e), \nabla_{\theta_e} L(\theta_l, \theta_e), \nabla_{\theta_l} L(\theta_l, \theta_e)$, these terms are the linear combination of $\mathcal{O}(T^2)$ computational complexity terms, so their computational complexities are also $\mathcal{O}(T^2)$.

If we directly compute the terms instead of approximating, use the expression of $\nabla_{\theta_l} f(\theta_l, \theta_e) = E^{\pi^{\theta_l, \theta_e}}[\sum_{t=0}^{T-1} \gamma^t[(\mu_l(s^t, a^t) - \mu_l(s^t))(J_{ed}(s^t, a^t) + J_{ei}(s^t, a^t))^T]] + \mu_l(\pi^{\theta_l, \theta_e})$ as an example, $\mu_l(s)$ is inside an expectation from $t = 0$ to $T - 1$, we need to sum up $\mu_l(s)$ for $T$ times. As a result, the computational complexity for $\nabla_{\theta_l} f(\theta_l, \theta_e)$ is $\mathcal{O}(T^3)$. Analogously, $\nabla_{\theta_e} f(\theta_l, \theta_e), \nabla^2_{\theta_l \theta_e} L(\theta_l, \theta_e), \nabla^2_{\theta_e \theta_e} L(\theta_l, \theta_e)$ are all same.

Back to SPSA, more policies need to be found compared to directly compute the gradient. For soft q learning (Haarnoja et al., 2017), we can find that for each epoch, there are $T$ iterations for $t$ ($t$ from 0 to $T - 1$). In each $t$, we need $\mathcal{O}(T)$ to compute the parameters for both single-agent and multi-agent cases. Therefore the computational cost for each epoch is $\mathcal{O}(T^2)$ and the overall computational cost is $\mathcal{O}(eT^2)$ where $e$ is the total number of epochs. The computational cost of the multi-agent RL is dominated by $\mathcal{O}(T^2)$. As a result, the computational complexity of SPSA is dominated by $\mathcal{O}(T^2)$. Since the calculation of the hypergradient requires second-order gradients (Lemma 2), directly calculating the hypergradient is dominated by $\mathcal{O}(T^3)$.

## A.10 EXPERIMENT DETAIL

The details of the experiments are shown in this section. All Python3 codes are run on a Windows 10 desktop with 13th Gen Intel(R) Core(TM) i7-13700KF CPU and 32 GB of RAM. For each combination of algorithms and environments, we run 10 times to calculate mean values and standard deviations at each iteration. Then the calculated mean values and standard deviations are plotted as shown in Section 8 figures.

### A.10.1 MPE

The state, action, and observation spaces for the adversary and good agents are continuous. For the adversary, it can observe the relative distance to the landmarks and the good agents, therefore the observation of the adversary is $o_a = [p_{l1} - p_a, p_{l2} - p_a, p_{g1} - p_a, p_{g2} - p_a]$ where $p_{l1}$ is the position of the first landmark, $p_{l2}$ is the position of the second landmark, $p_{g1}$ is the position of the first good agent, $p_{g2}$ is the position of the second good agent. For each good agent, it can observe the relative distance to the target landmark, the landmarks, the adversary, and another good agent, therefore the observations for two good agents are $o_{g1} = [p_{tl} - p_{g1}, p_{l1} - p_{g1}, p_{l2} - p_{g1}, p_a - p_{g1}, p_{g2} - p_{g1}]$ and $o_{g2} = [p_{tl} - p_{g2}, p_{l1} - p_{g2}, p_{l2} - p_{g2}, p_a - p_{g2}, p_{g1} - p_{g2}]$ where $p_{tl}$ is the position of the target landmark. The actions of the adversary and the good agents are the velocities between $0$ and $1$ in four directions (left, right, down, up). Two good agents share the same return, which is rewarded based on the minimum distance of any agent to the target landmark and is penalized based on the distance between the adversary and the target landmark, therefore the reward of good agents is $r_g = -\min(\|p_{tl} - p_{g1}\|_2, \|p_{t2} - p_{g1}\|_2) + \|p_{tl} - p_a\|_2$. The reward of the adversary is based on the distance to the target of the adversary, therefore $r_a = -\|p_{ta} - p_a\|_2$, where $p_{ta}$ is the position of the adversary's target. In our simulation, we consider observations as states of the MG, the distance $-\min(\|p_{tl} - p_{g1}\|_2, \|p_{t2} - p_{g1}\|_2)$ as the adversary-independent reward function, and the distance $\|p_{tl} - p_a\|_2$ as the feedback received by the good agents.

### A.10.2 HUMAN-ROBOT INTERACTION

In the simulation, the policies for the human and the robot are calculated through Multi-Agent Deep Deterministic Policy Gradient (MADDPG) (Lowe et al., 2017). During the training process, noises are added to the action to increase the exploration. Once the policies are generated, the further calculations use deterministic policies.

The state of the robot is its location $s_r = (x_r, y_r) \in \mathbb{R}^2$, the action of the robot includes the horizontal and vertical velocities and defined as $a_r = (v_{rx}, v_{ry})$, where $v_{rx} \in [-0.1, 0.1], v_{ry} \in [-0.1, 0.1]$. Similarly, the state of the human is $s_h = (x_h, y_h) \in \mathbb{R}^2$, the action of the human is $a_h = (v_{hx}, v_{hy})$, where $v_{hx} \in [-0.1, 0.1], v_{hy} \in [-0.1, 0.1]$. At each time step $t$, the robot chooses the action $a_r(t)$ based on the current joint state $(s_r(t), s_h(t))$ and moves to the next state $x_r(t+1) = x_r(t) + v_{rx}(t), y_r(t+1) = y_r(t) + v_{ry}(t)$. Analogously, the motion dynamics of the human is given by $x_h(t+1) = x_h(t) + v_{hx}(t), y_h(t+1) = y_h(t) + v_{hy}(t)$. In the experiment, the robot starts from an initial state $s_r(0) \in [-0.25, 0.25] \times [-0.4, 0.5]$ and aims to reach a circle goal region whose center is at $(0, 0.5)$ with radius $0.05$. The human starts from $s_h(0) \in [0.4, 0.5] \times [-0.25, 0.25]$ and aims to reach a circle goal region whose center is at $(-0.5, 0)$ with radius $0.05$. Both the robot and the human are penalized when a collision happens.

### A.10.3 SECURITY

There are 8 nodes and 10 edges. Each node represents a machine and each edge represents an exploit between two nodes. The decision-making of the defender and the attacker is modeled as an MG. The state $s \in \{0, 1\}^8$, represents the condition of each node where the value 1 means the current node is compromised by the attacker, and the value 0 is vice versa. In each action pair, the attacker chooses one edge to attack and the defender chooses one edge to block. Suppose the attack chooses to attack the edge $\{i, j\}$. If the node $i$ is already compromised and the defender does not block this edge, there is a probability for node $j$ to be compromised. For other situations, the node $j$ keeps clean. Each edge has a cost for the attacker to utilize and a cost for the defender to block. The attack receives a reward when it successfully compromises a new node. The net reward of the attacker for each state-action pair is the sum of the reward and the cost. For the defender, the expert-dependent

reward is the opposite of the attacker's reward and the expert-independent reward is the cost to block edges.

For the security simulation, the attack graph is randomly generated. We use Q-learning to find the policies for the attacker and the defender. During the training process, the attacker and the defender have a 70% possibility to choose between the best action with 60% possibility and the second best action with 40% possibility. Otherwise, the attacker and the defender randomly choose one action from the action space. When we exploit the learned policies, the attacker and the defender choose between the best action with 60% possibility and the second best action with 40% possibility.

### A.11 Limitation

Currently, the GSIIRL is based on the fully observed MG. However, in some cases, the learner can not observe all information about the environment. Therefore, the limitation of the GSIIRL is not considering the partially observed situation and we will develop an advanced algorithm to solve this limitation in the future.

