# OpenReview forum: "Learn from Interactions: General-Sum Interactive Inverse Reinforcement Learning"
_ICLR.cc/2025/Conference — Submitted to ICLR 2025_

### Official Review · Reviewer_865b · 2024-10-27

**Soundness:** 3
**Presentation:** 2
**Contribution:** 3
**Rating:** 6
**Confidence:** 2

**Summary:**

This paper considers inverse reinforcement learning (IRL) under the multi-agent setting, which aims to extend beyond the fully cooperative and fully competitive rewards in prior work. The authors propose a bi-level optimization problem and an IRL algorithm. Numerical results confirm the effectiveness of the proposed methodology.

**Strengths:**

1. The paper has a clear motivation and provides examples to show the commonality of the proposed setting.
2. The proposed algorithm attains comparable performance to the baseline without assessing demonstrations generated by ground truth rewards.

**Weaknesses:**

As multiple NEs may exist for a general-sum game, the proposed algorithm requires the expert and the learner to agree on the same NE (lines 4 and 10).

**Questions:**

1. Is it possible to also learn the expert's reward $r_e$ using $D_{r_{ld}^{\theta_l},r_e}$?
2. Does Theorem 2 guarantee the convergence to an NE?

---

> ### Author Response · Authors · 2024-11-24
>
> Thank you for your detailed and insightful review. Discussing these points will help us improve our work.
>
> **W1**:*As multiple NEs may exist for a general-sum game, the proposed algorithm requires the expert and the learner to agree on the same NE (lines 4 and 10).*
>
> **Answer**:The MGs in line 4 and line 10 are different, the MGs in line 4 are $(S, A, P, T, r_{ld}^{\theta_l}+r_{li},r_{e}, \gamma)$ and the MGs in line 10 are $(S, A, P, T, r_{ld}^{\theta_l}+r_{li},r_{e}^{\theta_e}, \gamma)$. The MG in line 10 is the approximation of the MG in line 4. In this paper, we assume the resulting policy from MARL is NE. These two NEs are different since they correspond to different reward functions ($r_{e}$ and $r_{e}^{\theta_e}$). For each MG we calculate NE through MARL once and then sample trajectories based on the calculated NE.
>
> **Q1**:*Is it possible to also learn the expert's reward $r_e$ through  using $D_{r_{ld}^{\theta_l},r_e}$*
>
> **Answer**:We learn the expert's reward $r_e$  from $D_{r_{ld}^{\theta_l},r_e}$. The only information related to $r_e$ that the learner can observe is $D_{r_{ld}^{\theta_l},r_e}$. With given $D_{r_{ld}^{\theta_l},r_e}$, we utilized the maximum likelihood IRL algorithm to learn  $r_e$ in our lower-level optimization problem.
>
> We do not guarantee to learn the ground truth $r_e$ of the expert. This is still a challenge for the whole IRL area. We add a Corollary 2.1 to the paper which proves that the cumulative reward difference between the learned expert policy and the ground truth expert policy decreases at a rate $\mathcal{O}(\frac{1}{\sqrt[4]{K}})$, even the learned expert's reward function is not identical to the ground truth reward function.
>
> **Q2**: *Does Theorem 2 guarantee the convergence to an NE?*
>
> **Answer**: *Theorem 2 does not guarantees the convergence to an optimal NE because the theorem 2 proves the learner's reward function converges to a stationary point of the non-convex upper-level objective function. If this stationary point is the optimal point. The learned learner's reward function converges to the optimal learner's reward function. With the learned learner's reward function and learned expert's reward function, we use MARL to get the joint policy. As we assume the policy calculated by MARL algorithms are NEs in this paper. This joint policy is NE and includes the learned learner policy and the learned expert policy. As the learner policy is soft bellman policy, it is a continuous function to the learned learner's reward function. Therefore, when the learned learner's reward function converges to the optimal learner's reward function,  the learned learner policy converges to the optimal learner policy [D1]. According to the Corollary 2.1,  the cumulative reward of the the learned expert policy converges to the ground truth expert policy.
>
> [D1] Rudin, W. (1964). Principles of mathematical analysis (Vol. 3). New York: McGraw-hill.*

---

### Official Review · Reviewer_aDCY · 2024-10-31

**Soundness:** 2
**Presentation:** 2
**Contribution:** 2
**Rating:** 3
**Confidence:** 3

**Summary:**

This paper proposes a stochastic bilevel method to tackle the problem of multi agent inverse reinforcement learning.
The authors manage to prove some convergence guarantees and conclude the paper with some experiments in simple settings.

**Strengths:**

Multi Agent IRL is an important open problem.

The author ran some experiments motivating their method.

**Weaknesses:**

The setting does not seem the standard inverse reinforcement learning setting where the learner does not access any reward.
In addition, the fact that the reward $r^{ld}$ can be evaluated along the learned trajectory makes not true the sttaement that the single agent case can be recovered if the learner does not affect the transitions and the reward of the expert. Therefore, overall I do not think that this method can be labeled as inverse reinforcement learning.

The theory is rather week in my view. In particular, how does a guarantee on the gradient norm transfer on the guarantees of the learner policy. It would be interesting to have a result saying that the learner policy converges to the best response to the expert but this is not shown in the paper.

In addition, in RL it is important to quantify not only the convergence rate but also the dependence on the number of states and actions. This is not evident in the theorem statement provided by this work.

Finally, the experiments are in my opinion carried out in only toy environment.

**Questions:**

How does the result in Theorem 2 translate in terms of guarantees for the suboptimality of the recovered policy measure against the expert policy with respect to the true reward function ?

How does the bound in Theorem 2 depend on the number of states/actions in tabular MDPs or on the complexity of the reward class in the function approximation case ?

---

> ### Author Response · Authors · 2024-11-24
>
> Thank you for your detailed and insightful review. Discussing these points will help us improve our work.
>
> **W1**: *The setting does not seem the standard inverse reinforcement learning setting where the learner does not access any reward. In addition, the fact that the reward
>  can be evaluated along the learned trajectory makes not true the statement that the single agent case can be recovered if the learner does not affect the transitions and the reward of the expert. Therefore, overall I do not think that this method can be labeled as inverse reinforcement learning.*
>
> **Answer**: This paper studies interactive IRL, which is different from classic IRL. In the classic IRL, the learner only passively observes the experts and learns from demonstrations in isolation from the experts. In interactive IRL [C1-C3], the learner actively interacts with the expert and the interactions influence the demonstrations of the expert. That is, the learner is not only an observer but also a participant. The expert will generate different demonstrations according to different learner's reward functions.  We can not directly apply classic IRL methods for our interactive IRL problem. For any learner's reward function, we can sample the demonstrations of the expert corresponding to the current learner's reward function. In this situation, the current learner's reward function is fixed (learner's reward function parameter $\theta_l$ is fixed), we can apply IRL to learn the expert's reward function $r_e^{\theta_e^*(\theta_l)}$ corresponding to the current learner's reward function. The policy corresponding to the $r_e^{\theta_e^*(\theta_l)}$ will mimic the behavior of the expert corresponding to the ground truth expert's reward function.
>
>  [C1]Büning, T. K., George, A. M., and Dimitrakakis, C. (2022, June). Interactive inverse reinforcement learning for cooperative games. In International Conference on Machine Learning (pp. 2393-2413). PMLR.
>
>  [C2]Hadfield-Menell, D., Russell, S. J., Abbeel, P., and Dragan, A. (2016). Cooperative inverse reinforcement learning. Advances in neural information processing systems, 29.
>
>  [C3]Zhang, X., Zhang, K., Miehling, E., and Basar, T. (2019). Non-cooperative inverse reinforcement learning. Advances in neural information processing systems, 32.
>
> **W2**:*The theory is rather week in my view. In particular, how does a guarantee on the gradient norm transfer on the guarantees of the learner policy. It would be interesting to have a result saying that the learner policy converges to the best response to the expert but this is not shown in the paper.*
>
> **Answer**:Theorem 2 focuses on the convergence of the reward function.  We add a Corollary 2.1 to the paper which proves that the cumulative reward difference between the learned expert policy and the ground truth expert policy decreases at a rate $\mathcal{O}(\frac{1}{\sqrt[4]{K}})$. In this paper, we assume the resulting policy from MARL is NE. Papers [C4-C6] introduce some MARL algorithms which approach NE. Given any reward functions of the learner of the expert, we can get the NE through MARL. When the learner's reward function converges,  the learner learns a joint policy which includes the learned learner policy and the learned expert policy. Since the joint policy is NE, we can say that the learned learner policy is the best response to the learned expert policy where the cumulative reward of learned expert policy converges to the cumulative reward of the ground truth expert policy.
>
> [C4]Prasad, H. L., LA, P., and Bhatnagar, S. (2015, May). Two-timescale algorithms for learning Nash equilibria in general-sum stochastic games. In Proceedings of the 2015 International Conference on Autonomous Agents and Multiagent Systems (pp. 1371-1379).
>
> [C5]Bowling, M. (2000, June). Convergence problems of general-sum multiagent reinforcement learning. In ICML (pp. 89-94).
>
> [C6] Hu, J., and Wellman, M. P. (2003). Nash Q-learning for general-sum stochastic games. Journal of machine learning research, 4(Nov), 1039-1069.

---

> ### Author Response · Authors · 2024-11-24
>
> **W3**:*In addition, in RL it is important to quantify not only the convergence rate but also the dependence on the number of states and actions. This is not evident in the theorem statement provided by this work.*
>
> **Answer**:Yes, in RL, it is important to consider the dimension (number) of the state and the action for convergence. However, existing works on IRL, e.g. [C7-C10] focus on the convergence of reward functions with respect to the iteration number or the number of trajectory samples. There are too many factors influence the convergence rate of the IRL. For example, the method used for learning the policy, the action space and the state space is continuous or discrete, the number of trajectories sampled during each iteration, the optimization problem is convex or non-convex, the difficulty of computing gradients and etc.
>
> [C7]Zeng, S., Li, C., Garcia, A., and Hong, M. (2022). Maximum-likelihood inverse reinforcement learning with finite-time guarantees. Advances in Neural Information Processing Systems, 35, 10122-10135.
>
> [C8]Liu, S., and Zhu, M. (2024). Learning multi-agent behaviors from distributed and streaming demonstrations. Advances in Neural Information Processing Systems, 36.
>
> [C9]Radanovic, G., Devidze, R., Parkes, D., and Singla, A. (2019, May). Learning to collaborate in markov decision processes. In International Conference on Machine Learning (pp. 5261-5270). PMLR.
>
> [C10]Goktas, D., Greenwald, A., Zhao, S., Koppel, A., and Ganesh, S. Efficient Inverse Multiagent Learning. In The Twelfth International Conference on Learning Representations.
>
> **W4**:*Finally, the experiments are in my opinion carried out in only toy environment.*
>
> **Answer**: In IRL, half cheetah, swimmer, and walker in MuJoCo are standard environments for experiments. In half cheetah, the action and state dimensions are 5 and 17, respectively. In swimmer, the action and state dimensions are 2 and 8, respectively. In walker, the action and state dimensions are 6 and 17, respectively. In our MPE experiment, the action and state dimensions are 10 and 28, respectively. As a result, our experiment has larger action and state dimensions than those of the aforementioned standard environments for IRL.
>
> **Q1**:*How does the result in Theorem 2 translate in terms of guarantees for the suboptimality of the recovered policy measure against the expert policy with respect to the true reward function ?*
>
> **Answer**:We add a Corollary 2.1 to the paper which proves that the cumulative reward difference between the learned expert policy and the ground truth expert policy decreases at a rate $\mathcal{O}(\frac{1}{\sqrt[4]{K}})$.  The convergence of the learner policy and the expert policy can also be visualized through the experiment. According to Figure 2, Figure 3(a), Figure 3(b), and the middle and right figures in Figure 4, we show that the cumulative rewards of the learner and the expert converge to the oracle. In the experiment, the MARL uses the ground truth reward functions of the learner and the expert to calculate the ground truth policy and then uses the ground truth policy to calculate the cumulative reward with ground truth reward functions. We use the results of MARL as the oracle. The cumulative reward of the GSIIRL is calculated through the learned policy and the ground truth reward functions.

---

> ### Author Response · Authors · 2024-11-24
>
> **Q2**:*How does the bound in Theorem 2 depend on the number of states/actions in tabular MDPs or on the complexity of the reward class in the function approximation case ?*
>
> **Answer**:As we focus on the convergence rate with respect to the iteration number or the number of trajectories, the bound does not include the number of states/actions. In the GSIIRL, we need methods such as MARL to generate policies to sample trajectories. Different MARL algorithms will lead to different values of bound with respect to the number of states/actions. For example, Nash Q learning [C11] is $\mathcal{O}(n|S||A|^n)$ where $n$ is the number of the agent, $|S|$ is the number of states, and $|A|$ is the number of actions. The algorithm in [C12] is $\mathcal{O}(|S||A|^n)$. Similarly, different reward classes will lead to different overall complexities with respect to the complexities of calculating the gradient of the reward function. If we use the neural network for the reward function, the complexity is $\mathcal{O}(d_S \times d_A)$  where $d_S$ is the state dimension and $d_A$ is the action dimension. The complexity of the neural network depends on the input dimension. As the input of the reward function is states and actions, the complexity of the neural network reward function depends on the dimension of the state and the dimension of the action.
>
> [C11]  Hu, J., and Wellman, M. P. (2003). Nash Q-learning for general-sum stochastic games. Journal of machine learning research, 4(Nov), 1039-1069.
>
> [C12]Liu, Q., Yu, T., Bai, Y., and Jin, C. (2021, July). A sharp analysis of model-based reinforcement learning with self-play. In International Conference on Machine Learning (pp. 7001-7010). PMLR.

---

> > ### Comment · Reviewer_aDCY · 2024-11-26
> >
> > Dear authors,
> >
> > Thanks for your response.
> >
> > I maintain my original score since I still do not find convincing the arguments that the authors used to say that their environment are more difficult than MuJoCo and because they did not state a theorem proving rigorously the convergence to a Nash equilibrium.
> >
> > Corollary 2.1 does not imply convergence to Nash.
> >
> > Moreover, the dependence on states and actions is very important to be quantified and the authors didn't address this concern in their revision.

---

### Official Review · Reviewer_18t7 · 2024-11-03

**Soundness:** 1
**Presentation:** 1
**Contribution:** 1
**Rating:** 3
**Confidence:** 3

**Summary:**

This paper proposes a novel framework for general-sum interactive inverse reinforcement learning (IRL) where a learner aims to infer the reward function of an expert through interaction rather than passive observation. The authors introduce a stochastic bi-level optimization approach, embodied in their General-Sum Interactive Inverse Reinforcement Learning (GSIIRL) algorithm, which features a double-loop structure: the inner loop learns the expert’s reward function, while the outer loop refines the learner’s policy. Unlike traditional IRL methods focused on fully cooperative or competitive setups, this approach allows for complex interaction dynamics, where the learner’s reward can be decomposed into expert-dependent and expert-independent parts.

**Strengths:**

I am afraid it is difficult to find a strength in its current state.

**Weaknesses:**

Please see comments for details on weaknesses and what must be improved.
# The Writing

The writing is in bad shape and must be improved drastically.

- There are language errors such as "Until now, IRL has been applied to
different areas." where the use of "until" is wrong. Also, there are many vague statements such as "Current state-of-the-arts (Ziebart
et al., 2008; Abbeel & Ng, 2004; Ziebart et al., 2010; Bagnell et al., 2006; Finn et al., 2016) solve this problem with different approaches." that does not describe any of these approaches and do not add any content to the paper. These kinds of vague and generic statements must be immediately concretised afterwards, or not be made at all.

- "The aforementioned works are only focused on a single expert and a single learner. In the real world,
many scenarios have multiple experts interacting with each other in the environment." -- Here it s not clear whether you are talking about a dataset of demonstrations for a single-agent task performed by different experts, or a multi-agent task. Also, what are these real-world scenarios? You should give an example immediately after this statement, otherwise again it is hand-wavy and vague.

- The whole of Introduction is extremely convoluted, filled with vague statements, and difficult to read. This needs to be re-written entirely with a good flow and with proper scientific writing. Right now, it reads like the draft of a draft. By the time I reach line 74, I still have no idea what the paper is about. It is also extremely difficult to assess the paper's relation to previous work because of how convoluted the introduction is. Is this multi-agent IRL? Is this single-agent multi-expert IRL?

- The Related Works should be a section, not a \paragraph inside the Introduction. In the Related Works, you do not need to include all the different things bi-level optimization is used for. Those are not related works to your paper. Your related works should contain related inverse RL papers and how your work differs from them. Right now, this paragraph is a long and unnecessary background on bilevel optimization.

# The Novelty and Problem Setting

I believe the paper's problem statement is wrong, in the sense that, it does not require the use of inverse RL, and IRL is not justified in any other way either.

- In the motivating example, the robot's goal is to compute an optimal policy with respect to a reward function that depends on the human. So this seems like it is a two-agent setting where there is a robot and a human. In Section 2, it is unclear where the inverse RL is needed. Any model-free multi-agent RL algorithm that can solve a Markov Game can be used to compute a good policy for robot here, because the robot gets to observe its rewards even if it does not know the function. This is standard model-free RL.

- The paper must cite the "Efficient Inverse Multiagent Learning" work (https://openreview.net/forum?id=JzvIWvC9MG) and make similar comparisons to their work and works mentioned in Table b. Specifically, the bilevel optimization method presented here looks extremely similar to the minmax formulation presented there, and also presented in other inverse MARL works. However, I reiterate that it is incredibly difficult to assess the paper's novelty due to the writing.

- I also do not understand what the empirical results tell us. In your setting, both the robot (i.e. learner) and the human (i.e. expert) actually can observe their rewards when a joint action is executed. Then, for the robot, there is no reason to do inverse RL. Standard MARL is enough.

- The point above makes me think that there is a crucial misunderstanding here. Inverse RL is not applied to settings where simply the reward function is not _known_ by the agent. As long as the rewards can be observed by the agent, standard RL (i.e. model-free) is enough. We use IRL for settings where the agent cannot even observe rewards, as in, once robot takes an action it does not receive any reward feedback, thus even the feedback itself must be learned. This is not the case in your work.

**Questions:**

Please feel free to address any points mentioned under the Weaknesses.

---

> ### Author Response · Authors · 2024-11-24
>
> Thank you for your detailed and insightful review. Discussing these points will help us improve our work. We have substantially revised the first two sections to further clarify the problem we study and why our problem cannot be solved by existing methods, e.g., IRL and multi-agent RL (MARL). The updated version of the paper has been uploaded.
>
> **W1**:*In the motivating example, the robot's goal is to compute an optimal policy with respect to a reward function that depends on the human. So this seems like it is a two-agent setting where there is a robot and a human. In Section 2, it is unclear where the inverse RL is needed. Any model-free multi-agent RL algorithm that can solve a Markov Game can be used to compute a good policy for robot here, because the robot gets to observe its rewards even if it does not know the function. This is standard model-free RL.*
>
> **Answer**: Yes, the motivating example is a two-agent setting where the agents are the robot and the human. The goal of the robot is to learn the expert's reward function. Referring to the seminal work [B1], `` the reward function is the most succinct, robust, and transferable definition of the task''. The benefits of learning the human's reward include facilitating the robot's policy learning and learning a transferable human model. When the transition probabilities of a MG substantially change, the policy needs to be re-trained, but the reward function could be the same. During the interaction, the only thing containing the information about the human's reward function available to the robot is the trajectories (demonstrations) of the human. Therefore, learning the human's reward function is an IRL problem. Though model-free MARL can be used to compute the policy, it can not be utilized to learn the reward function of the expert.
>
> [B1]Ng, A. Y., and Russell, S. (2000, June). Algorithms for inverse reinforcement learning. In Icml (Vol. 1, No. 2, p. 2).
>
> **W2**: *The paper must cite the "Efficient Inverse Multiagent Learning" work (https://openreview.net/forum?id=JzvIWvC9MG) and make similar comparisons to their work and works mentioned in Table b. Specifically, the bilevel optimization method presented here looks extremely similar to the minmax formulation presented there, and also presented in other inverse MARL works. However, I reiterate that it is incredibly difficult to assess the paper's novelty due to the writing.*
>
> **Answer**: We cite the mentioned paper as an example of the fully competitive interactive IRL problem and it is a special of our interactive IRL problem. Assuming the expert-independent reward function of the learner as 0 and the expert-dependent reward function of the learner is opposite to the expert's reward function. Both the expert and the learner aim to maximize their own cumulative reward and equivalently minimize the cumulative reward of the other one. As a result, from the learner's perspective, our bi-level problem becomes the lower-level learning the expert's reward function by minimizing the cumulative reward of the learner and the upper-level learning the learner reward function by maximizing the cumulative reward of the learner. Since the objective functions of the lower-level optimizing problem and the upper-level optimizing problem are identical, our bi-level optimization problem reduces to the minmax formulation in the provided reference. In our problem, we consider the relationship between the reward functions of the learner and the expert to be arbitrary and this is our novelty.
>
> **W3**: *I also do not understand what the empirical results tell us. In your setting, both the robot (i.e. learner) and the human (i.e. expert) actually can observe their rewards when a joint action is executed. Then, for the robot, there is no reason to do inverse RL. Standard MARL is enough.*
>
> **Answer**: We agree that, if the robot only wants to learn the policy of the robot, model-free MARL is enough. Referring to the answer for weakness 1, in our problem, the goal of the robot is learning the human's reward function. Learning the human's reward function is achieved through IRL.   Model-free MARL can not be used to learn the human's reward function.

---

> ### Author Response · Authors · 2024-11-24
>
> **W4**: *The point above makes me think that there is a crucial misunderstanding here. Inverse RL is not applied to settings where simply the reward function is not known by the agent. As long as the rewards can be observed by the agent, standard RL (i.e. model-free) is enough. We use IRL for settings where the agent cannot even observe rewards, as in, once robot takes an action it does not receive any reward feedback, thus even the feedback itself must be learned. This is not the case in your work.*
>
> **Answer**: Referring to the answer for weakness 1, the learner aims to learn the reward function of the expert. The learner can observe the reward for itself, but it can not observe the reward of the expert. Since the learner can only observe the trajectories of the expert, learning the reward of the expert is an IRL problem and we apply IRL to learn the reward function of the expert.

---

> ### Comment · Reviewer_18t7 · 2024-11-26
>
> From the motivating example you provide: "This goal of the robot is captured by the reward function $h(p_r , p_h, v_r , v_h).$ The function $h$ is unknown to the robot but the robot can measure the value of $h(p_r , p_h, v_r , v_h)$ given
> any locations $p_r , p_h$ and velocities $v_r , v_h$." -- So this setting is clearly not the IRL setting. In inverse RL, _we do not have observations of the reward function._ So the robot should not have any measurement or observation of their reward function $h(p_r , p_h, v_r , v_h).$ If an agent can observe their own rewards, they can just use model-free RL to optimize for those rewards. But then I see there at the end you state the robot's reward function is "$−||g_r −p_r||_2 +h(p_r , p_h, v_r , v_h)$." Then the robots goal is not captured just by $h$? Also have't you just called $h$ the robot's reward function earlier? The goal should be represented by the reward function. All terms are observed by the robot here, because $h$ is also observed. Then the robot gets reward observations in the and, therefore not IRL. Specifically, I do not see why the "robot" has to learn the expert's reward function, if they can observe the $h$ which is the only part that depends on the expert in their reward function.
>
> Now, we can argue over this back and forth, but at the end even with all things considered, even the point above indicates a big writing problem. The clarity of the paper needs major improvement, and I am still not convinced this is the IRL setting. I do not believe this clears the bar for ICLR, regardless of the point on IRL.

---

### Official Review · Reviewer_FWZk · 2024-11-04

**Soundness:** 3
**Presentation:** 2
**Contribution:** 3
**Rating:** 6
**Confidence:** 4

**Summary:**

This paper introduces the General-Sum Interactive Inverse Reinforcement Learning (GSIIRL) framework, which uses inverse reinforcement learning to enhance the learner´s policy optimization process in a multi-agent setting. In GSIIRL, a learner agent interacts with an expert to infer the expert´s reward function, using this insight to adapt its policy within a general-sum environment without assuming equilibrium conditions.

**Strengths:**

1. **Structured Learner-Expert Framework:** GSIIRL´s learner-expert setup, where the learner infers the expert´s reward and adjusts its own policy, provides a unique structure within multi-agent IRL. This interaction-focused approach enables the learner to balance its intrinsic rewards with its understanding of the expert´s objectives in a general-sum setting, which is an important extension to IRL in interactive environments.
2. **Flexibility in Non-Equilibrium Scenarios:** Unlike methods like MA-AIRL, which often rely on equilibrium assumptions, GSIIRL operates without this requirement, making it well-suited to dynamic, real-world environments where equilibrium conditions may not be guaranteed.
3. **Theoretical Analysis:** The paper provides theoretical convergence guarantees and error-bound analysis, strengthening the reliability of the GSIIRL framework.

**Weaknesses:**

1. **Simplistic Expert Reward Structure:** GSIIRL currently assumes a single, static reward function for the expert, which may limit its flexibility in more complex environments where the expert´s objectives could adapt dynamically in response to the learner´s actions. This static assumption places GSIIRL closer to frameworks like MA-AIRL, which also assume fixed rewards, potentially reducing the depth of interactions in dynamic multi-agent scenarios.
2. **Assumption of Observability:** As recognized by the authors, the assumption of full access to state-action pairs limits GSIIRL´s practicality, especially when compared to frameworks like MA-AIRL, which are designed to operate under partial observability.

**Questions:**

1. Could the authors elaborate at a high level on whether the GSIIRL framework could be extended to operate under partial observability? Additionally, how would such an extension theoretically impact the framework´s reward inference and policy optimization, especially regarding convergence guarantees and error bounds?
2. Could the authors discuss the rationale behind using a single, static reward function for the expert, rather than a structure with both intrinsic and learner-responsive components? How might incorporating a more flexible, adaptive reward structure for the expert impact the framework´s effectiveness in dynamic multi-agent environments?

---

> ### Author Response · Authors · 2024-11-24
>
> Thank you for your detailed and insightful review. Discussing these points will help us improve our work. Since two weaknesses are also included in two questions, we integrate the answers to the weaknesses into those to the questions.
>
> **Q1**: *Could the authors elaborate at a high level on whether the GSIIRL framework could be extended to operate under partial observability? Additionally, how would such an extension theoretically impact the framework´s reward inference and policy optimization, especially regarding convergence guarantees and error bounds?*
>
> **Answer**:
> GSIIRL can be extended to partial observability situations. In specific, the MG under partial observability [A1] becomes $(S, A, Z, P, T, r_{ld}+r_{li},r_{e}, O, \gamma)$ where $Z= Z_l \times Z_e$ is the finite set of observations, $O(z;a,s)$ denotes the probability density of perceiving observation $z \in Z$ when taking action $a$ and arriving in state $s$. Then we need to consider the belief $b$ as the probability distribution over states and $b(s)$ as the probability of the state is $s$ at the current time step. The policy becomes $\pi(a|b)$. The trajectories observed by the learner become the collection of observations $z$ and actions $a$ instead of states $s$ and actions $a$. Through the collection of $z$ and $a$, a collection of $b$ can be calculated through equation (3) in [A1].
>
> For the GSIIRL, the inputs for reward functions become $b$ and $a$. Although $b$ is continuous, we can make this change because GSIIRL considers continuous state space. For policy learning, we can still use  MARL but the complexity will increase for POMDP[A2]. For the reward learning, we can still use maximum likelihood IRL. For each $b$, we need to consider the effect of all $z$ and $a$ before the current time step. The data of $b$ is more noisy than the the state data. Therefore, we need more trajectories and the complexity of the reward learning will also increase. New proofs are needed for convergence guarantees and error bounds in the POMDP setting.
>
>
>
> [A1]Choi, J. D., and Kim, K. E. (2011). Inverse reinforcement learning in partially observable environments. Journal of Machine Learning Research, 12, 691-730.
>
> [A2]Egorov, M. (2015). Deep reinforcement learning with pomdps. Tech. Rep.(Technical Report, Stanford University, 2015), Tech. Rep.
>
>
> **Q2**: *Could the authors discuss the rationale behind using a single, static reward function for the expert, rather than a structure with both intrinsic and learner-responsive components? How might incorporating a more flexible, adaptive reward structure for the expert impact the framework´s effectiveness in dynamic multi-agent environments?*
>
> **Answer**: In GSIIRL, we design the reward function as a neural network, and the dimension of the neural network could be manually set up. When we increase the dimension of the neural network, it has a higher possibility of capturing the task of the expert and the learner-responsive components. One possible solution for extending GSIIRL to the dynamic environments is to set up the reward function as time-varying [A3]. Therefore, in GSIIRl we can consider the expert-dependent reward parameter $\theta_l^t$ and the expert reward parameter $\theta_e^t$ are time-varying. For now, each trajectory has $T$ time steps. At each time step $t$, we can calculate one pair of  $\theta_l^t$ and $\theta_e^t$. Therefore, we will get $T$ pairs of reward parameters in total. We need more trajectories to include the information about how the reward function changes corresponding to $t$ and $T$ becomes a significant factor in the convergence rate.  The exact change to the convergence rate and the error bound need further rigorous proof.
>
>
>
>
> [A3]Ashwood, Z., Jha, A., and Pillow, J. W. (2022). Dynamic inverse reinforcement learning for characterizing animal behavior. Advances in neural information processing systems, 35, 29663-29676.

---

### Author Response · Authors · 2024-11-24

Hi reviewers,

Thank you for your detailed and insightful review. Discussing these points will help us improve our work. We have substantially revised the first two sections to further clarify the problem we study and why our problem cannot be solved by existing methods, e.g., IRL and multi-agent RL (MARL). The updated version of the paper has been uploaded.

---

### Meta-Review · Area_Chair_CFd1 · 2024-12-20

**Metareview:**

The paper tackles the problem of multi-agent inverse reinforcement learning (IRL), proposing a bi-level optimization framework. While the topic is relevant and timely, there are significant concerns about the paper’s alignment with classical IRL, as the learner can observe rewards along the trajectory, which diminishes the need for IRL. Several reviewers raised doubts about the theoretical guarantees, particularly regarding convergence to a Nash equilibrium. Theoretical results were seen as insufficient, with Corollary 2.1 failing to establish convergence to Nash and the dependence on the number of states and actions left unaddressed.

Empirical results were limited to toy environments, and the clarity of the paper was criticized, particularly in explaining the relationship to prior work and the novelty of the approach. Although the proposed method shows comparable performance to baselines, the lack of rigorous proof and the unclear theoretical foundations make it difficult to assess its broader applicability and impact.

In conclusion, while the paper tackles an important problem, it does not yet meet the standards for publication due to theoretical and empirical limitations, as well as writing and clarity issues.

**Additional Comments On Reviewer Discussion:**

After the authors' rebuttals, the reviewers maintained their original scores. They still questioned the method’s classification as IRL, as the learner observes rewards, which diminishes the need for IRL. The theoretical guarantees remained insufficient, particularly regarding convergence to Nash equilibrium, and the empirical results in toy environments did not convince reviewers of the method's broader applicability. The writing was still criticized for being unclear and convoluted. Overall, the reviewers felt the paper did not meet ICLR's standards due to these unresolved issues.

---

### Decision · Program_Chairs · 2025-01-22

Reject